# Pink Noise Is All You Need: Colored Noise Exploration in Deep Reinforcement Learning

**Onno Eberhard**[1] **Jakob Hollenstein**[2,1] **Cristina Pinneri**[3,1] **Georg Martius**[1]

[1]Max Planck Institute for Intelligent Systems, Tübingen, Germany [2]Universität Innsbruck
[3]Max Planck ETH Center for Learning Systems
`{firstname.lastname}@tuebingen.mpg.de`

## Abstract

In off-policy deep reinforcement learning with continuous action spaces, exploration is often implemented by injecting action noise into the action selection process. Popular algorithms based on stochastic policies, such as SAC or MPO, inject white noise by sampling actions from uncorrelated Gaussian distributions. In many tasks, however, white noise does not provide sufficient exploration, and temporally correlated noise is used instead. A common choice is Ornstein-Uhlenbeck (OU) noise, which is closely related to Brownian motion (red noise). Both red noise and white noise belong to the broad family of *colored noise*. In this work, we perform a comprehensive experimental evaluation on MPO and SAC to explore the effectiveness of other colors of noise as action noise. We find that pink noise, which is halfway between white and red noise, significantly outperforms white noise, OU noise, and other alternatives on a wide range of environments. Thus, we recommend it as the default choice for action noise in continuous control.

## 1 Introduction

Exploration is vitally important in reinforcement learning (RL) to find unknown high reward regions in the state space. This is especially challenging in continuous control settings, such as robotics, because it is often necessary to coordinate behavior over many steps to reach a sufficiently different state. The simplest exploration method is to use action noise, which adds small random perturbations to the policy's actions. In off-policy algorithms, where the exploratory behavioral policy does not need to match the target policy, action noise may be drawn from any random process. If the policy is deterministic, as in DDPG (Lillicrap et al., 2016) and TD3 (Fujimoto et al., 2018), action noise is typically white noise (drawn from temporally uncorrelated Gaussian distributions) or Ornstein-Uhlenbeck (OU) noise, and is added to the policy's actions. In algorithms where the policy is stochastic, such as SAC (Haarnoja et al., 2018) or MPO (Abdolmaleki et al., 2018), the action sampling itself introduces randomness. As the sampling noise is typically uncorrelated over time, these algorithms effectively employ a scale-modulated version of additive white noise, where the noise scale varies for different states.

Figure 1: Trajectories of pure noise agents on a bounded integrator environment (Sec. 6). White action noise (left) does not reach far in this environment, and it would not be able to collect a sparse reward at the edges: it *explores locally*. 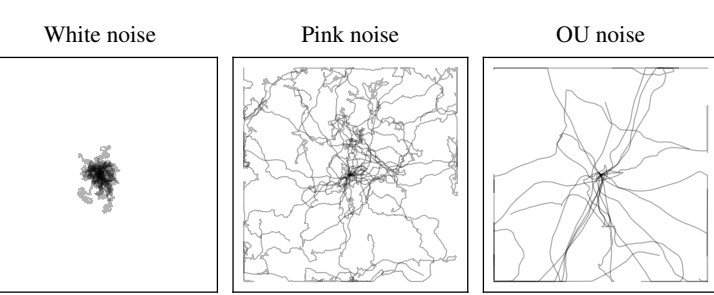 OU noise (right) only *explores globally* and gets stuck at the edges. Pink noise (center) provides a balance of local and global exploration, and covers the state space more uniformly than the other two.

In many cases, white noise exploration is not sufficient to reach relevant states. Both MPO and SAC have severe problems with certain simple tasks like MountainCar because of inadequate exploration. As in TD3 or DDPG, the off-policy nature of these algorithms makes it possible to replace the white noise process, which is implicitly used for action sampling, by a different random process. The effectiveness of temporal correlation in the action selection has been noted before (e.g. Osband et al., 2016) and is illustrated in Fig. 1, where the exploration behavior of white noise (uncorrelated) is compared to that of noises with intermediate (pink noise) and strong (OU noise) temporal correlation on a simple integrator environment (more on this in Sec. 6). Using highly correlated noise, such as OU noise, can yield sufficient exploration to deal with these hard cases, but it also introduces a different problem: strongly off-policy trajectories. Too much exploration is not beneficial for learning a good policy, as the on-policy state-visitation distribution must be covered during training to make statistical learning possible. Thus, a typical approach is to use white noise by default, and alternatives like OU noise only when necessary. In this work, our goal is to find a better strategy, by considering noises with intermediate temporal correlation, in the hope that these work well both on environments where white noise is enough, and on those which require increased exploration.

To this end, we investigate the effectiveness of colored noise as action noise in deep RL. Colored noise is a general family of temporally correlated noise processes with a parameter $\beta$ to control the correlation strength. It generalizes white noise ($\beta = 0$) and Brownian motion (red noise, $\beta = 2$), which is closely related to OU noise. We find that average performance across a broad range of environments can be increased significantly by using colored action noise with intermediate temporal correlation ($0 < \beta < 2$). In particular, we find *pink noise* ($\beta = 1$) to be an excellent default choice. Interestingly, pink noise has also been observed in the movement of humans: the slight swaying of still-standing subjects, as well as the temporal deviations of musicians from the beat, have both been measured to exhibit temporal correlations in accord with pink noise (Duarte & Zatsiorsky, 2001; Hennig et al., 2011).

Our work contributes a comprehensive experimental evaluation of various action noise types on MPO and SAC. We find that pink noise has not only the best average performance across our selection of environments, but that in $80\%$ of cases it is not outperformed by any other noise type. We also find that pink noise performs on par with an oracle that tunes the noise type to an environment, while white and OU noise perform at $50\%$ and $25\%$ between the worst noise type selection and the oracle, respectively. To investigate whether there are even better noise strategies, we test a color-schedule that goes from globally exploring red noise to locally exploring white noise over the course of training, as well as a bandit method to automatically tune the noise color to maximize rollout returns. Both methods, though they significantly improve average performance when compared to white and OU noise, are nevertheless significantly outperformed by pink noise. In addition to the results of our experiments, we attempt to explain why pink noise works so well as a default choice, by constructing environments with simplified dynamics and analyzing the different behaviors of pink, white and OU noise. *Our recommendation is to switch from the current default of white noise to pink noise.*

## 2 BACKGROUND & RELATED WORK

Reinforcement learning (RL) has achieved impressive results, particularly in the discrete control setting, such as achieving human-level performance in Atari games with DQN (Mnih et al., 2015) or mastering the game of Go (Silver et al., 2016) by using deep networks as function approximators. In this paper, we are concerned with the continuous control setting, which is especially appropriate in robotics. In continuous action spaces, it is typically intractable to choose actions by optimizing a value function over the action space. This makes many deep RL methods designed for discrete control, such as DQN, not applicable. Instead, researchers have developed policy search methods (e.g. Williams, 1992; Silver et al., 2014), which directly parameterize a policy. These methods can be divided into on-policy algorithms, such as TRPO (Schulman et al., 2015) and PPO (Schulman et al., 2017), and off-policy algorithms such as DDPG, TD3, SAC and MPO.

All these algorithms have to address the problem of exploration, which is fundamental to RL: in order to improve policy performance, agents need to explore new behaviors while still learning to act optimally. One idea to address exploration is to add a novelty bonus to the reward (e.g. Thrun, 1992). In deep RL, this can be done by applying a bonus based on sample density (Tang et al., 2017) or prediction error (Burda et al., 2019). Another method to encourage exploration is to take inspiration from bandit methods like Thompson sampling (e.g. Russo et al., 2018), and act optimistically with

respect to the uncertainty in the Q-function (Osband et al., 2016). The simplest strategy, however, is to randomly perturb either the policy parameters (Plappert et al., 2018; Mania et al., 2018), or the actions themselves. This can be done by randomly sampling a function for each episode that deterministically alters the action selection (Raffin & Stulp, 2020), by learning correlations between action dimensions and state space dimensions to induce increasing excitation in the environment (Schumacher et al., 2022), or by learning an action prior from task-agnostic data (Bagatella et al., 2022).

In this work, we consider the simplest and most common form of exploration in continuous control: action noise. Action noise can be either explicitly added to the policy, or implicitly, by randomly sampling actions from a stochastic policy. The most common form of action noise is white noise, which typically comes from sampling from independent Gaussian distributions at every time step. Apart from white action noise, Lillicrap et al. (2016) successfully used temporally correlated Ornstein-Uhlenbeck noise, and Pinneri et al. (2020) achieved improvements in model predictive control by utilizing colored noise. Inspired by this success, in this work we investigate the effectiveness of colored action noise in the context of model-free RL, specifically on MPO and SAC.

## 3 METHOD

In this paper, we investigate exploration using action noise. In algorithms like DDPG and TD3, where the learned policy $\mu$ is deterministic, action noise is simply added to the policy:

$$a_t = \mu(s_t) + \sigma\varepsilon_t, \tag{1}$$

where $\varepsilon_{1:T} = (\varepsilon_1, \ldots, \varepsilon_T)$ is sampled from a random process, and $\sigma$ is a scale parameter. If $\varepsilon_t$ is sampled independently at every time step, e.g. from a Gaussian distribution, then $\varepsilon_{1:T}$ is called *white noise* (WN). This is the prevailing choice of action noise, though it is also common to use time-correlated Ornstein-Uhlenbeck noise ($\varepsilon_{1:T} \sim \mathrm{OU}_T$) (Uhlenbeck & Ornstein, 1930).

Algorithms which parameterize a stochastic policy, such as SAC and MPO, also use action noise. In continuous action spaces, the most common policy distribution is a diagonal Gaussian, represented by the functions $\mu(s_t)$ and $\sigma(s_t)$: $a_t \sim \mathcal{N}(\mu(s_t), \mathrm{diag}(\sigma(s_t))^2)$. This can equivalently be written as

$$a_t = \mu(s_t) + \sigma(s_t) \odot \varepsilon_t, \tag{2}$$

where $\varepsilon_t \sim \mathcal{N}(0, I)$. In this case, the action noise $\varepsilon_{1:T}$ is again Gaussian white noise, which is scale-modulated by the function $\sigma$.

White noise is not correlated over time ($\mathrm{cov}[\varepsilon_t, \varepsilon_{t'}] = 0$). In some environments, this leads to very slow exploration, which in turn leads to inadequate state space coverage, leaving high reward regions undiscovered. Thus, it is often beneficial to use action noise with temporal correlation ($\mathrm{cov}[\varepsilon_t, \varepsilon_{t'}] > 0$), like Ornstein-Uhlenbeck (OU) noise. OU noise was recommended as the default choice for DDPG, and has been shown to lead to a significant increase in state space coverage (Hollenstein et al., 2022). OU noise is defined by the stochastic differential equation (SDE)

$$\dot{\varepsilon}_t = -\theta\varepsilon_t + \sigma\eta_t, \tag{3}$$

where $\eta_t$ is a white noise process. If $\theta = 0$, then this equation defines integrated white noise, also called Brownian motion. Brownian motion is temporally correlated, but cannot be used as action noise if generated in this way, because its variance increases unboundedly over time, violating the action space limits. This problem is addressed by setting $\theta > 0$ (a typical choice is $\theta = 0.15$), which bounds the variance. More details about OU noise and Brownian motion can be found in Sec. A.

A broad family of temporally correlated noises is given by *colored noise*, which generalizes both white noise and Brownian motion (in this context called *red noise*).

**Definition 1** (Colored noise). A stochastic process is called colored noise with color parameter $\beta$, if signals $\varepsilon(t)$ drawn from it have the property $|\hat{\varepsilon}(f)|^2 \propto f^{-\beta}$, where $\hat{\varepsilon}(f) = \mathcal{F}[\varepsilon(t)](f)$ denotes the Fourier transform of $\varepsilon(t)$ ($f$ is the frequency) and $|\hat{\varepsilon}(f)|^2$ is called the power spectral density (PSD).

The color parameter $\beta$ controls the amount of temporal correlation in the signal. The PSDs of colored noise with different $\beta$ are shown in Fig. A.2. If $\beta = 0$, then the signal is uncorrelated, and the PSD is flat, meaning that all frequencies are equally represented. This noise is called white noise in analogy to light, where a signal with equal power on all visible frequencies is perceived as white.

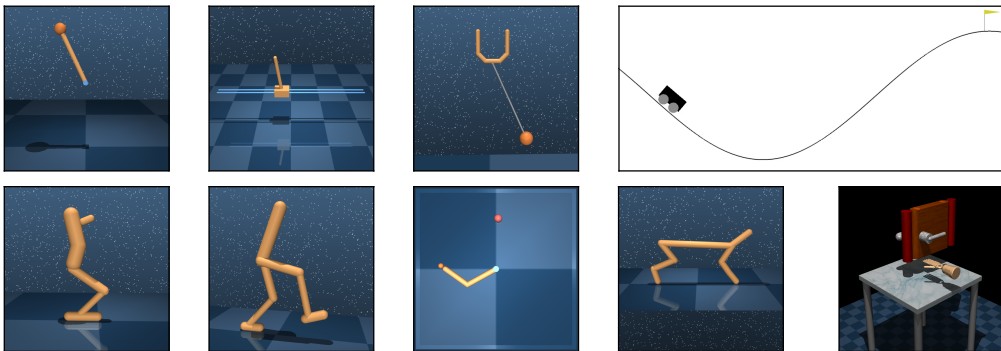

Figure 2: The environments we use: Pendulum, CartPole (balance + swingup tasks), Ball-In-Cup, MountainCar, Hopper, Walker, Reacher, Cheetah, Door. See Sec. C for more details. Images partly taken from Tassa et al. (2018) with permission.

Red noise ($\beta = 2$) is named so, because it has more weight on lower frequencies, which in light corresponds to the red part of the spectrum. Gaussian colored noise with constant variance can be efficiently generated, and the complete noise signal for an episode can be sampled at once, to be used as action noise according to Equations (1) and (2). If generated like this, which we denote by $\varepsilon_{1:T} \sim \mathrm{CN}_T(\beta)$ (more details in Sec. A), white noise is identical to independently sampling from a Gaussian distribution at every time step. Red noise ($\mathrm{CN}_T(2)$) is very similar to OU noise with the default setting $\theta = 0.15$, as both are essentially Brownian motion with bounded variance (see Fig. A.2).[1] By setting $0 < \beta < 2$, colored noise allows us to search for a better default action noise type with intermediate temporal correlation between white and red noise. One special case is *pink noise*, which is defined by $\beta = 1$.

## 4 IS PINK NOISE ALL YOU NEED?

Fujimoto et al. (2018) found that the type of action noise (white or OU) in general does not influence the performance of TD3. In contrast to this, Hollenstein et al. (2022) found that the noise type does have an influence, but that the impact of this choice, as well as which noise type is preferable, depends entirely on the environment. We start by confirming these latter results[2], and compare white and OU action noise with a selection of colored action noises ($\beta \in [0, 2]$), in terms of the achieved performance. In all of our experiments we use MPO and SAC, relying on the implementations by Pardo (2020) and Raffin et al. (2021), respectively. We found that both algorithms significantly outperform TD3 across tasks, and thus only briefly discuss TD3 in Sec. B.1. Since the optimality of an action noise type depends on the environment, we perform experiments on a diverse set of 10 different tasks taken from the DeepMind Control Suite (Tassa et al., 2018), OpenAI Gym (Brockman et al., 2016), and the Adroit hand suite (Rajeswaran et al., 2018). These environments are shown in Fig. 2 and are described in more detail in Sec. C. We report results on some additional tasks in Sec. G.

To evaluate the performance of a training procedure (which always lasts $10^6$ environment interactions), we run 5 evaluation rollouts every $10^4$ interactions. We then report the performance as the mean return of all these evaluation rollouts. Since this performance is related to the area under the learning curve, it is a measure combining both the final policy performance, and the sample efficiency of an algorithm. More detailed results, including learning curves and an analysis of the final policy performance, can be found in Sections B.2 and H.

### 4.1 DOES THE NOISE TYPE MATTER?

To assess the importance of the choice of action noise, we evaluate the performances achieved by SAC and MPO when using white noise, OU noise and colored noise as action noise (where

---

[1]Other settings of $\theta$ for OU noise, which are less similar to red noise, are discussed in Sec. A. In the main text we only consider the default setting of $\theta = 0.15$.

[2]We also confirm the former results (see Sec. B.1), but find that colored noise (especially pink noise) outperforms both white and OU noise on TD3.

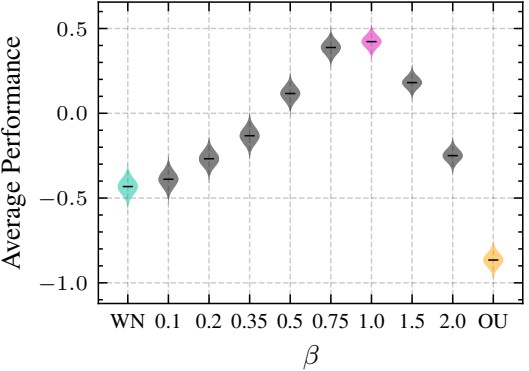

| Environment | Best noise | $p$ | Pink? |
|---|---|---|---|
| Pendulum | 2.0 | **0.01** | ✗ |
| Cartpole (b.) | 1.0 (Pink) | — | ✓ |
| Cartpole (s.) | 1.0 (Pink) | — | ✓ |
| Ball-In-Cup | 0.75 | 0.88 | ✓ |
| MountainCar | 2.0 | 0.59 | ✓ |
| Hopper | 1.0 (Pink) | — | ✓ |
| Walker | 0.5 | 0.36 | ✓ |
| Reacher | White noise | **0.02** | ✗ |
| Cheetah | 0.75 | 0.62 | ✓ |
| Door | 0.75 | 0.65 | ✓ |

Figure 3: Bootstrap distributions for the expected average performance of MPO and SAC using different action noise types (details in Sec. B.2). Highlighted are white noise (WN), pink noise ($\beta = 1$), and Ornstein-Uhlenbeck noise (OU).

Table 1: A Welch $t$-test reveals that the performance difference between pink noise and the *best noise* is only significant in two out of ten environments. The rightmost column answers whether pink noise performs equally well as the best noise type.

$\beta \in \{0.1, 0.2, 0.35, 0.5, 0.75, 1, 1.5, 2\}$), on the benchmark environments shown in Fig. 2. We repeat all learning runs with 20 different seeds, resulting in a total of $20 \times 10 \times 2 \times 10 = 4000$ experiments[3], each one reporting a single scalar performance. To control for the influence of the algorithm and environment on the performance of a particular noise type, we group all results by algorithm and task, and normalize them to zero mean and unit variance. We then calculate a noise type's *average performance*: the normalized performance of all runs using this noise type, averaged across algorithms and environments. In Fig. 3, bootstrap distributions for the expected average performances are shown, generated using the 20 random seeds available per task and algorithm (more details in Sec. B.2). It can be seen that the noise type indeed matters for performance. A clear preference for pink noise ($\beta = 1$) becomes visible, which considerably outperforms white noise and OU noise across tasks. In Sec. B.2, this performance difference can be seen on the corresponding learning curves, where we compare white, OU and pink noise.

Achieving the best average performance across environments is not the same as being the best performing option on each individual environment. This begs the question of when pink noise (the best general option) also performs as good as an environment's best noise type. We perform a Welch $t$-test for each environment to check whether the expected difference between the performances of pink noise and the task-specific best noise[4] is significant. The results are listed in Table 1. Although pink noise only achieves the highest mean across seeds in three of the ten tasks, the statistical analysis reveals that the difference between pink noise and the best noise type is only significant in two out of ten cases. In other words, in the tested environments, pink noise performs on par with the best choice of noise type in 80% of cases! What about the two environments, Pendulum and Reacher, where pink noise is outperformed by other noise types? On Pendulum, pink noise, on average, achieves 83% of the performance of red noise ($\beta = 2$). In contrast, white noise only achieves 39% of the performance of red noise (OU performs similarly to red noise). On Reacher, pink noise achieves 99% of white noise's performance, while OU noise achieves only 76%. So, even on the few environments where pink noise is outperformed significantly, it is clearly preferable as a default over white noise and OU noise. These results indicate that *pink noise* seems to be *all you need*.

## 4.2 IS PINK NOISE A GOOD DEFAULT?

The best performance on a given environment is always achieved with the task-specific best action noise type. It would be nice to always use this best noise type, but it is often unpractical to run a large hyperparameter search to find it, especially when including many possible colors $\beta$. It is common therefore, to stick to a "default" choice, which is typically white noise. In the previous section, we saw that pink noise is a better default choice than white noise, but it is still unclear whether this is

---

[3]seeds × tasks × algorithms × noise types (WN + OU + 8 colors)

[4]To compute the *best* noise type, we normalize out the contribution of the algorithm (similarly to the *average performance*) and take the mean performance over random seeds on each environment.

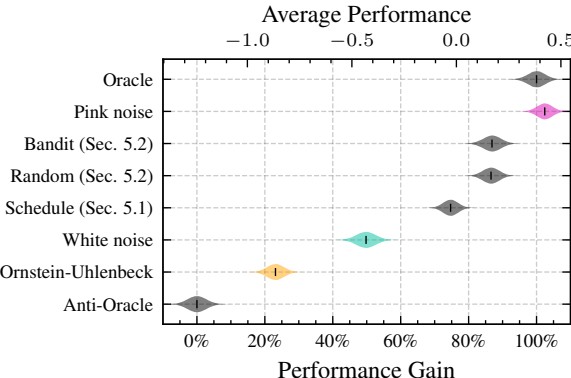

Figure 4: Bootstrap distributions for the expected average performances of all methods we discuss in this paper. Highlighted are again white noise and OU noise (the popular options), as well as pink noise (our suggestion). While OU noise and white noise only achieve about $25\%$ and $50\%$ of the possible performance gain of an oracle method, pink noise performs equally to the oracle! Pink noise is also not outperformed by a color-scheduling method (Sec. 5.1), nor by a bandit algorithm (Sec. 5.2).

enough, or if a hyperparameter search might be needed for good performance. In this section, we will analyze how much performance is lost by sticking to a default value of white noise, pink noise, or OU noise, compared to using the task-specific best noise type.

We choose the task-specific best noise type via an "oracle" method, which can be thought of as a very extensive grid search: an environment's best noise type is selected by looking at the results of all noise types, and choosing the best performing option across 10 seeds. By also doing an "anti-oracle" experiment, which selects the worst noise type on each environment (i.e. the most unlucky pick of noise types possible), we can define a new "performance gain" measure, which might be easier to interpret than the average performance in the previous section.[5] The performance gain of a noise type specifies where its average performance falls between the anti-oracle's performance ($0\%$) and the oracle's performance ($100\%$). Figure 4 presents the performance gains of using white noise, Ornstein-Uhlenbeck noise and pink noise as a default for all environments.

By always sticking to Ornstein-Uhlenbeck noise, only about $25\%$ of the highest possible performance gain is achieved, and the resulting performance would be closer to using the anti-oracle. By using white noise instead, we already achieve a performance gain of over $50\%$. However, picking pink noise does not appear to sacrifice any performance compared to the oracle![6] The gain achieved by switching from white noise as the default to pink noise is both considerable and significant, and we recommend switching to *pink noise as the default action noise*.

## 5 ALL THE COLORS OF THE RAINBOW

We have found that pink noise is the best default action noise over a broad range of environments. There are still some environments, however, where pink noise is outperformed by other noise types, specifically by white and red noise on the Reacher and Pendulum tasks, respectively. This indicates that there may not exist a single noise type which performs best on all environments. However, this consideration is only valid if the noise is kept constant over the course of training. If we instead try a different approach, and choose the noise type separately for each rollout, we may find a strategy that is outperformed nowhere. In this section we discuss two such non-constant methods, which differ in the way a rollout's noise type is selected: a color-schedule going from $\beta = 2$ to $\beta = 0$, and a bandit approach with the intention of finding the optimal color for an environment.

### 5.1 IS COLOR-SCHEDULING BETTER THAN PINK NOISE?

To find a more effective exploration method than pink noise, it is helpful to understand why learning in the Pendulum and Reacher environments works better with other noise types. The Pendulum environment is underactuated and requires a gradual build-up of momentum by slowly swinging back and forth. Strongly correlated action noise, such as red noise, makes this behavior much more likely. The Reacher task, on the other hand, has neither a particularly large state space, nor does it exhibit

---

[5]The (anti-)oracle is evaluated on the 10 seeds not used for noise type selection to avoid sampling bias. We repeat this by selecting (evaluating) once on the first (latter) 10 seeds, and once on the latter (first) 10 seeds.

[6]It looks as if pink noise is even exceeding the oracle's performance. This difference is not statistically significant and is due to the oracle only having access to the 10 seeds not used for evaluation.

| Environment | P > S | P < S | Pink ≥ Schedule? | P > B | P < B | Pink ≥ Bandit? |
|---|---|---|---|---|---|---|
| Pendulum | | <0.01 | ✗ | | 0.73 | ✓ |
| Cartpole (b.) | <0.01 | | ✓ | 0.10 | | ✓ |
| Cartpole (s.) | 0.01 | | ✓ | <0.01 | | ✓ |
| Ball-In-Cup | <0.01 | | ✓ | 0.47 | | ✓ |
| MountainCar | | 0.15 | ✓ | 0.29 | | ✓ |
| Hopper | 0.05 | | ✓ | 0.15 | | ✓ |
| Walker | <0.01 | | ✓ | | 0.28 | ✓ |
| Reacher | <0.01 | | ✓ | | 0.50 | ✓ |
| Cheetah | 0.05 | | ✓ | 0.64 | | ✓ |
| Door | 0.15 | | ✓ | 0.27 | | ✓ |

Table 2: How does pink noise compare to a schedule and a bandit method? For each environment, we perform a Welch $t$-test to test for inequality of the performances of pink noise vs. the schedule/bandit method. The $p$-values are arranged to show which performance is higher. Pink noise performs significantly better than the schedule on most environments. Compared to the bandit algorithm, pink noise performs better overall, but on most environments the difference is not significant. In the "Pink ≥ ...?" columns, (✓) means that pink noise does not perform significantly worse than the alternative.

under-actuation, such that white noise is well suited to explore the space. Here, temporally correlated noise will only lead to off-policy trajectory data, thereby inhibiting learning. In general, strongly correlated noise leads to more global exploration, while uncorrelated noise explores more locally. We return to these ideas in Sec. 6, where we analyze the effects of high and low correlation on two simple environments.

Our method should work well on both of these environments, and on environments which require a mix of local and global exploration. Thus, we are looking for a strategy which balances local and global exploration. A simple idea to do this is a color-schedule: start with highly correlated red noise ($\beta = 2$) and then slowly decrease $\beta$ to white noise ($\beta = 0$) over the course of training. The rationale behind this strategy is that high-reward regions can be quickly discovered at the beginning of training when $\beta$ is large, while the trajectories get more on-policy over time, helping with environments like Reacher. Indeed, a similar approach that schedules the action noise *scale*, has been shown to work quite well (Hollenstein et al., 2022).

We implement a $\beta$-schedule, which linearly goes from $\beta = 2$ to $\beta = 0$, on MPO and SAC and repeat the experiment with 20 random seeds on all environments. Bootstrap distributions for the expected average performance across environments are shown in Fig. 4 (denoted by *Schedule*). The results indicate that the schedule is generally better than OU and white noise, but does not outperform pink noise. Indeed, pink noise is significantly better, as the confidence intervals do not overlap. If we take a more detailed look at the individual environments (Table 2), we see that thanks to the additional highly correlated noise, the schedule does outperform pink noise on the Pendulum environment, as expected. However, in all other environments pink noise either significantly outperforms the schedule or they perform on par, so our recommendation to use pink noise as a default remains.

## 5.2 IS BANDIT COLOR SELECTION BETTER THAN PINK NOISE?

The results in the previous section indicate that, while changing the noise type over the training process can improve performance, simply moving from globally exploring red noise to more locally exploring white noise does not outperform pink noise. Instead of trying to find a different schedule to fit all environments, in this section we consider an adaptive approach. By using a bandit algorithm to select the action noise color for each rollout on the basis of past rollout returns, it might be possible to find not only the general best noise for a given environment, but even to automatically adapt the noise to different stages of training. The bandit algorithm we use is based on Thompson sampling, the details are explained in Sec. D.

We use the rollout return itself as the bandit reward signal. The reasoning for this is that in environments where strong exploration is necessary (such as Pendulum and MountainCar), high return will only be achieved by strongly correlated actions. On the other hand, if environments do not require correlated actions, or a capable policy has been learned, the highest return should be achieved by the action noise which least disturbs the policy, i.e. noise with a low correlation.

As an additional baseline, we also perform an experiment where a color ($\beta$) is randomly selected for each rollout.[7] For both methods we use the same list of $\beta$ values as in Sec. 4.1 (incl. $\beta = 0$), and repeat the experiments with 20 random seeds. The results on MPO and SAC are shown in Fig. 4 (marked with *Bandit* and *Random*) and Table 2. It can be seen that the bandit method is again outperformed by pink noise. Indeed, a bootstrapping test yields a highly significant difference in expected average performance across environments ($p = 0.005$). Looking at the results on the individual tasks (Table 2), it seems like the bandit method does outperform pink noise on the two problematic environments (Pendulum and Reacher), however, this difference is not significant. A detailed comparison of the bandit and its random baseline can be found in Table D.1, which shows that neither of the two methods significantly outperforms the other on any environment. This indicates that, while there may be merit in changing the noise type over the training process, the rollout return appears to contain too little information to effectively guide the noise type selection. Thus, our recommendation to use pink noise as a default remains unchanged.

## 6  HOW DO ACTION NOISE AND ENVIRONMENT DYNAMICS INTERACT?

Why is pink noise such a good default noise type? In Sec. 5.1, we briefly discussed the concepts of local and global exploration, and hypothesized that the best exploration behavior provides a balance of the two, such that high reward regions will be found, while trajectories are not too off-policy. To analyze how different noise types behave, we will look at a simplified *bounded integrator* environment: a velocity-controlled 2D particle moving in a box (more details in Sec. F.2). If we control this particle purely by noise, we can analyze the exploration behavior in isolation of a policy. As a first test, we run 20 episodes of 1000 steps in an environment of size $250 \times 250$ with white noise, pink noise, and OU noise (all with unit variance, $x$- and $y$-velocity controlled independently). The resulting trajectories are shown in Fig. 1. It can be seen that pink noise provides the best combination of local and global exploration: it reaches the edges (unlike white noise), but does not get stuck there (unlike OU noise).

A good mix of local and global exploration gives rise to a more uniform state space coverage, as can be seen in Fig. 1. Thus, how well a noise type explores depends highly on the size of the environment: if the environment was much smaller, white noise would be enough to cover the space and pink noise trajectories would look similar to the OU trajectories shown here. On the other hand, if the environment were bigger, then pink noise would not reach the edges and OU noise would explore better. The uniformity of the state space coverage is measured by the entropy of the state-visitation distribution. We estimate the entropy induced by a noise type using a histogram density approximation: we partition the state space into a number of boxes ($50 \times 50 = 2500$ boxes), sample $10^4$ trajectories, and count the number of sampled points in each box.

Figure 5 shows the entropy achieved by white noise, OU noise and pink noise as a function of the environment size. The sizes are chosen to reflect the complete sensible range for episode lengths of 1000 steps, each with unit variance: from very small ($50 \times 50$) to very large ($2000 \times 2000$). Pink noise is not "special" in the sense that it performs best on all environments, as we already saw in the previous sections. However, it performs best on "medium scales", as determined by the episode length, and does not suffer from severe degradation in performance over the whole spectrum of sensible environments. If we do not know where on this spectrum a given environment lies, then pink noise is clearly a better default choice than white noise or OU noise!

Besides integrating actions, another common aspect of environment dynamics is oscillation. Oscillation dynamics are dominant in the Pendulum and MountainCar environments[8], but also relevant in other domains, like Ball-In-Cup, Cartpole, and Walker. To model these dynamics, we construct a second environment: a simple harmonic oscillator. This system is a frictionless 1-dimensional physical setup, in which a mass $m$ is attached to an ideal spring of stiffness $k$. The state space consists of the mass's position and velocity, and the action describes a force that is applied to the mass. The goal is to maximize the energy in the oscillator system (which is equivalent to maximizing the amplitude), similar to the MountainCar and Pendulum tasks, where this is necessary to collect the sparse reward.

The oscillator environment is parameterized by the resonant frequency $f$ of the system, which is fixed by setting the stiffness $k = 4\pi^2$ and the mass $m = 1/f^2$ (more details in Sec. F.1). Figure 5 shows

---

[7]This method would be roughly equivalent to the bandit method if we provided no bandit reward signal.

[8]See Sec. E for a simple method exploiting this property to solve MountainCar.

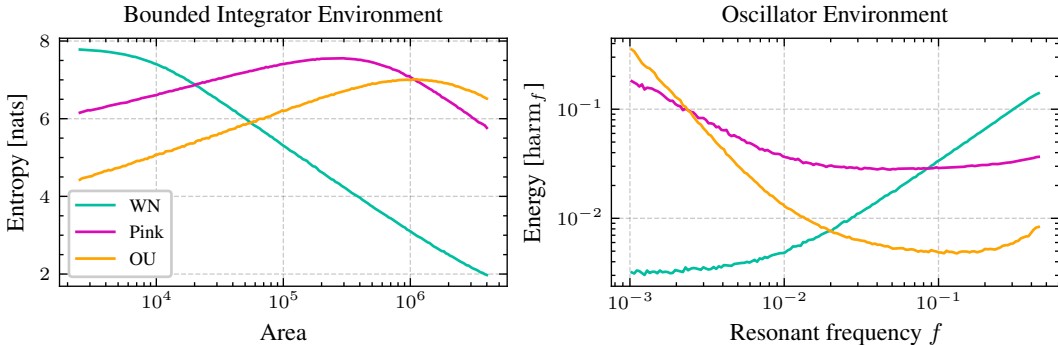

Figure 5: Pink noise strikes a favorable middle ground between white noise and Ornstein-Uhlenbeck noise on a wide range of environments. On both a bounded integrator environment parameterized by its size (left), and on a simple harmonic oscillator environment parameterized by its resonant frequency (right), it is much more general in terms of the range of parameters which yield good results, and performs well on the complete range of reasonable parameterizations. We argue that this quality is what makes it a good default.

the average energy in the oscillator system (over 1000 episodes of 1000 steps each) as a function of the resonant frequency $f$, which we vary from very low ($f = \frac{1}{1000}$, episode length = 1 period) to very high ($f = \frac{1}{2}$, Nyquist frequency), when driven by white noise, pink noise, and OU noise. The energy is measured relative to the average energy achieved by a sinusoidal excitation at the resonant frequency, denoted $\mathrm{harm}_f$. Even though this is a completely different setup to the bounded integrator, and we are using a very different performance metric, the two plots look remarkably similar. Again, this shows the power of pink noise as a default action noise: if we do not know the resonant frequency of the given environment, pink noise is the best choice.

These two environments (bounded integrator and oscillator) are rather simplistic. However, the dynamics of many real systems undoubtedly contain parts which resemble oscillations (when a spring or pendulum is present), single or double integration (when velocities/steps or forces/torques are translated into positions) or contact dynamics (such as the box in the bounded integrator). If an environment's dynamics are very complex, i.e. they contain many such individual parts, then the ideal action noise should score highly on each of these "sub-tasks". However, if all these individual parts have different parameters (like the environment size or resonant frequency above), it stands to reason that the best single action noise would be the one which is general enough to play well with all parameterizations, i.e. *pink noise*. On the flip side, the average performance in Fig. 3 over all environments may be interpreted as the performance over a very complicated environment, with the sub-tasks being the "actual" environments. This might explain why we see this curve: all sub-tasks have very different parameters, and require different action noises (as seen in Table 1), but pink noise is general enough to work well on all sub-tasks, and thus easily outperforms noise types like white noise or OU noise, which are only good on very specific environments (see Fig. 5).

## 7 CONCLUSION

In this work we performed a comprehensive experimental evaluation of colored noise as action noise in deep reinforcement learning for continuous control. We compared a variety of colored noises with the standard choices of white noise and Ornstein-Uhlenbeck noise, and found that pink noise outperformed all other noise types when averaged across a selection of standard benchmarks. Pink noise is only significantly outperformed by other noise types on two out of ten environments, and overall performs equally well to an oracle selection of the noise type. Additionally, we compared pink noise to more sophisticated methods that change the noise type over the course of training: a color-schedule, a bandit method, and a random selection scheme. No method outperforms pink noise, and our recommendation is to *use pink noise as the default action noise*. Finally, we studied the behaviors of pure noise agents on two simplified environments: a bounded integrator and a harmonic oscillator. The results showed that pink noise is much more general with respect to the environment parameterization than white noise and OU noise, which sheds some light on why it performs so well as the default choice.

ACKNOWLEDGMENTS

We want to thank Marco Bagatella, Sebastian Blaes, and Pierre Schumacher for helpful feedback on earlier revisions of this text, and the Max Planck ETH Center for Learning Systems for supporting Cristina Pinneri. Georg Martius is a member of the Machine Learning Cluster of Excellence, EXC number 2064/1 – Project number 390727645. We acknowledge the support from the German Federal Ministry of Education and Research (BMBF) through the Tübingen AI Center (FKZ: 01IS18039B).

REPRODUCIBILITY STATEMENT

We submitted our code as supplementary material, and provide a polished version of it online (`https://github.com/martius-lab/pink-noise-rl`). In an effort to ensure reproducibility, we took particular care in applying random seeds to all randomized parts of the algorithms, including the environments, algorithm initialization and training, as well as the action noise signals. In Sec. C, we provide details on the exact algorithms and environments we use. The hyperparameters for our own methods are included in the code submission. Additionally, our main experiments were all repeated with 20 different random seeds to eliminate statistical flukes.

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

# Supplementary Material

## PINK NOISE IS ALL YOU NEED

## A  COLORED NOISE AND ORNSTEIN-UHLENBECK NOISE

Colored noise has an interesting property that was not mentioned in the main text: integrating a colored noise signal with parameter $\beta$ again yields a colored noise signal, only with parameter $\beta + 2$. This stems from the property of the Fourier transform that an integration in the time domain corresponds to a multiplication with $(i2\pi f)^{-1}$ in the frequency domain. Let $v(t)$ be the original colored noise signal with $|\hat{v}(f)|^2 \propto f^{-\beta}$. Then the PSD of $x(t) = \int_0^t v(\tau) \, d\tau$ is

$$|\hat{x}(f)|^2 = \left| \mathcal{F}\left[ \int_0^t v(\tau) \, d\tau \right](f) \right|^2 = \left| \frac{1}{i2\pi f} \hat{v}(f) \right|^2 \propto f^{-2} |\hat{v}(f)|^2 \propto f^{-(\beta+2)}. \tag{4}$$

From this, and the definition of white noise as colored noise with $\beta = 0$, it follows that Brownian motion (integrated white noise) is also colored noise with parameter $\beta = 2$. In Fig. A.1, sampled signals of most of the noise types we use in this paper are shown, and in Fig. A.2, we plot the power spectral densities of some of these.

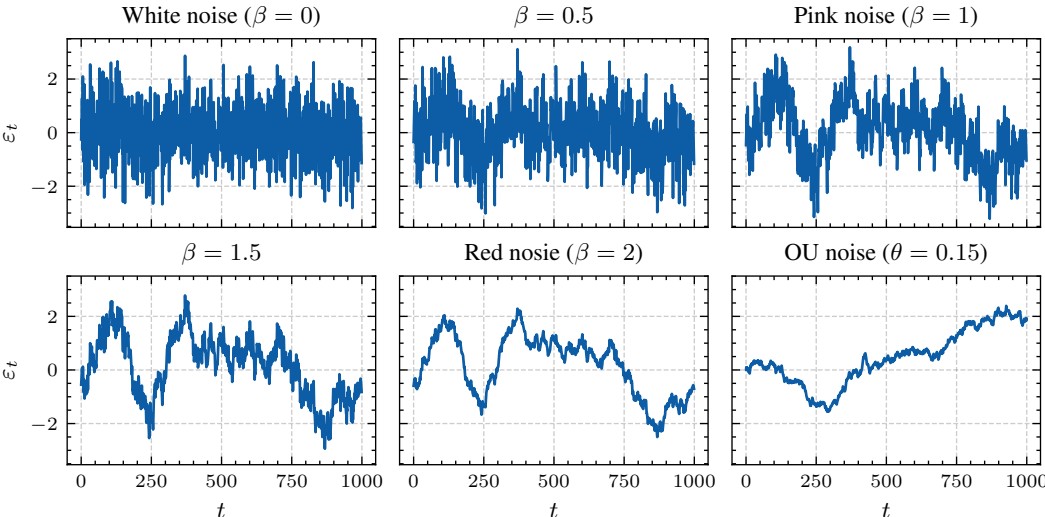

Figure A.1: Sampled signals from various action noise processes with noise scale $\sigma = 1$. The exception is OU noise, whose noise scale is adjusted such that $\mathrm{var}[\varepsilon_t] = 1$ (see discussion below).

We generate colored noise using the procedure described by Timmer & Koenig (1995), based on the Fast Fourier Transform (FFT) (Cooley & Tukey, 1965). This method is very efficient, as it only requires sampling a Gaussian signal in the frequency space (where the PSD is shaped), and then transforming it to the time domain via the FFT. In particular, this procedure is faster than sampling an Ornstein-Uhlenbeck signal (using the most common procedure, which we describe below). We use the colorednoise Python package (https://github.com/felixpatzelt/colorednoise) to sample colored noise signals, and always sample signals of the complete episode length (which we denote by $\varepsilon_{1:T} \sim \mathrm{CN}_T(\beta)$). The Python implementation contained a bug, which among other things made it so the generated "white noise" was correlated, and our fix of this bug is included as of version 2.1.0 of the package. Colored noise sampled according to this procedure is stationary and Gaussian: the signals are marginally identical to standard Gaussian distributions, i.e. $p(\varepsilon_t) = \mathcal{N}(\varepsilon_t \mid 0, 1)$. The only difference to white noise (independent Gaussian samples at every time step) is that they are temporally correlated: $p(\varepsilon_t, \varepsilon_{t'}) \neq p(\varepsilon_t) p(\varepsilon_{t'})$. This is shown empirically on the example of pink noise in Fig. A.3.

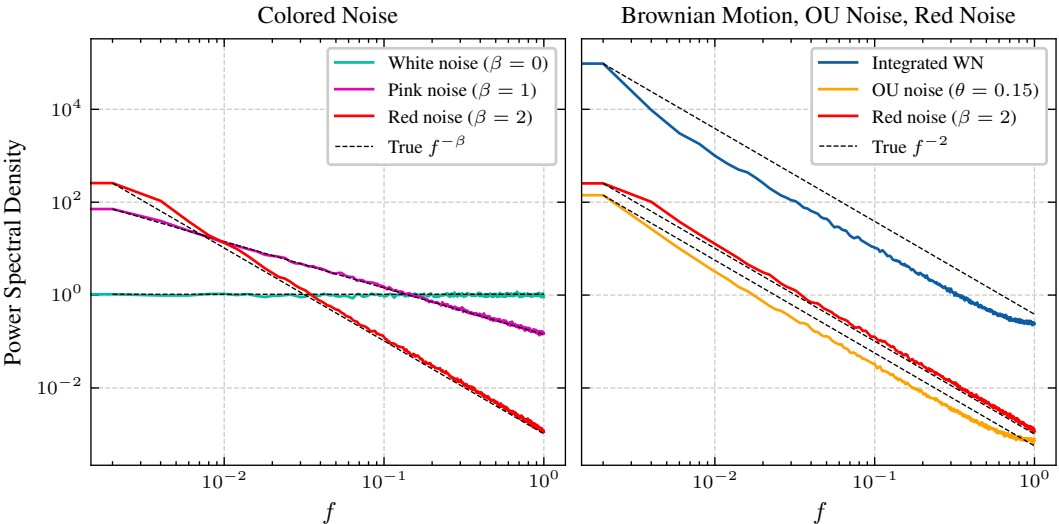

Figure A.2: Left: The power law trends can be seen in the PSDs of sampled colored noise signals. Right: Brownian motion, here generated by integrating white noise sampled from $\mathcal{N}(0,1)$, is compared to two related stationary noises: Ornstein-Uhlenbeck noise ($\theta = 0.15$), and red noise. The similarity between OU and red noise is visible. All signals are of length $T = 1000$.

## A.1    ORNSTEIN-UHLENBECK NOISE GENERATION AND VARIANCE CORRECTION

Also included in Fig. A.3 is Ornstein-Uhlenbeck (OU) noise. It can be seen that OU noise starts out as non-stationary but quickly converges to the same marginal distribution $p(\varepsilon_t) = \mathcal{N}(\varepsilon_t \mid 0, 1)$ as the other noise types. Important to note is that all these noise types are suitable for use as action noise only because they are (or quickly become) stationary, and hence their variance does not grow without bounds (contrary to that of Brownian motion). The property that all noise types have the same marginal distribution shows that our results are only due to a change in the *temporal correlation* of the action noise, not in the scale or shape of the distribution, as this is the same as of regular Gaussian white noise. To make sure that OU noise converges to a standard Gaussian marginal distribution we cannot use a noise scale of $\sigma = 1$, but have to correct it. Ornstein-Uhlenbeck noise can be defined by the stochastic differential equation

$$\mathrm{d}x_t = -\theta x_t \,\mathrm{d}t + \sigma \,\mathrm{d}w_t\,, \tag{5}$$

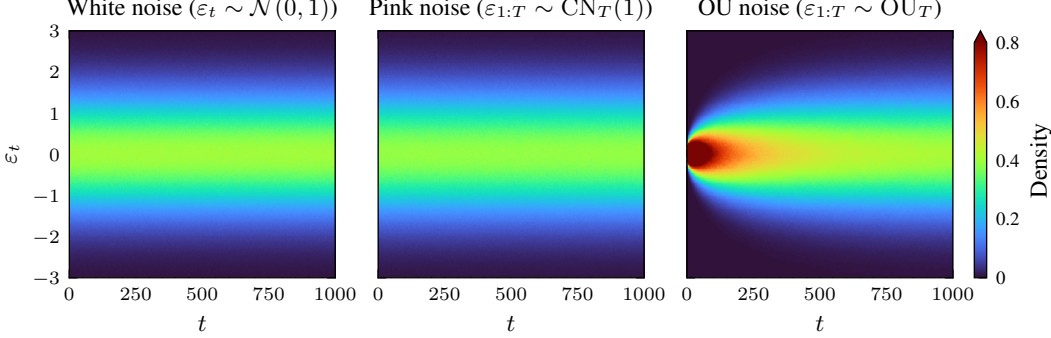

Figure A.3: The colored noise we use as action noise has the same marginal distribution as independent Gaussian samples. We sampled $3 \times 10^5$ action noise signals of length $T = 1000$ from each of the following random processes: independent Gaussian samples (white noise, left), pink noise (center), Ornstein-Uhlenbeck noise (right). At every time step $t$ we show a histogram density estimate over action noise values $\varepsilon_t$. This shows that our results are only due to the increased temporal correlation of the action noise signals, as the marginal distributions remain unchanged from white noise.

where $w_t$ is a Wiener process (integrated white noise with the property that $w(t) - w(t') \sim \mathcal{N}(0, t - t')$ for any $0 \leq t' < t$). This definition of Ornstein-Uhlenbeck noise is equivalent to the Langevin equation (3) in the main text, but is nicer to work with, as the white noise process $\eta_t$ is ill-defined as the derivative of the Wiener process. We sample OU noise signals by discretizing the equation above:

$$x[t + \Delta t] = x[t] - \theta x[t] \Delta t + \sigma \sqrt{\Delta t} \varepsilon, \tag{6}$$

where $\varepsilon \sim \mathcal{N}(0, 1)$. Denoting $x_t := x[t \Delta t]$ (with $x_{-1} = 0$) and $\varepsilon_t \sim \mathcal{N}(0, 1)$ for all $t \in \mathbb{N}_0$, it can be seen that

$$
\begin{aligned}
x_0 &= \sigma \sqrt{\Delta t} \varepsilon_0 \\
x_1 &= x_0 - \theta x_0 \Delta t + \sigma \sqrt{\Delta t} \varepsilon_1 \\
&= \sigma \sqrt{\Delta t}(1 - \theta \Delta t) \varepsilon_0 + \sigma \sqrt{\Delta t} \varepsilon_1 \\
x_2 &= \sigma \sqrt{\Delta t}(1 - \theta \Delta t)^2 \varepsilon_0 + \sigma \sqrt{\Delta t}(1 - \theta \Delta t) \varepsilon_1 + \sigma \sqrt{\Delta t} \varepsilon_2 \\
&\vdots \\
x_t &= \sigma \sqrt{\Delta t} \sum_{\tau=0}^{t} (1 - \theta \Delta t)^{t-\tau} \varepsilon_\tau.
\end{aligned}
$$

Thus, as a sum of zero-mean Gaussian distributions, the marginal distribution is:

$$
\begin{aligned}
p(x_t) &= \sigma \sqrt{\Delta t} \mathcal{N}\left(0, \sum_{\tau=0}^{t} ((1 - \theta \Delta t)^{t-\tau})^2\right) \\
&= \mathcal{N}\left(0, \sigma^2 \Delta t \sum_{\tau=0}^{t} (1 - \theta \Delta t)^{2\tau}\right).
\end{aligned}
$$

The variance of this distribution is a geometric series which converges as $t \to \infty$ if $(1 - \theta \Delta t)^2 < 1$, which holds if $0 < \theta \Delta t < 2$. It is interesting to note that if $\theta \Delta t = 1$, then Eq. (6) yields white noise, as it reduces to $x_t = \sigma \sqrt{\Delta t} \varepsilon$. On the other hand, if $\theta \Delta t = 0$, the equation describes integrated white noise (Brownian motion), which is known to have unbounded variance. If $1 < \theta \Delta t < 2$, then the signal exhibits negative temporal correlation, which follows from Eq. (6). If the geometric series converges, then the limiting variance is given by

$$\frac{\sigma^2 \Delta t}{1 - (1 - \theta \Delta t)^2}.$$

We can thus ensure a standard Gaussian marginal distribution (in the limit) by setting the noise scale to a "corrected" value of

$$\sigma = \sqrt{\frac{1 - (1 - \theta \Delta t)^2}{\Delta t}}, \tag{7}$$

which is how we set the OU noise scale throughout the paper to make the comparison with white and colored noise fair. In Fig. A.3, it can be seen that this limiting marginal distribution is reached fairly quickly. In Sec. B, we also report Ornstein-Uhlenbeck results with the more common choice of $\sigma = 1$, which we find to generally perform slightly worse (cf. Fig. B.1).

If the variance is corrected, then $\theta \Delta t$ is the only parameter of OU noise, such that e.g. $(\theta = 0.3, \Delta t = 1)$ is equivalent to $(\theta = 30, \Delta t = 0.01)$. This immediately follows by plugging Eq. (7) into Eq. (6), yielding

$$x_{t+1} = (1 - \theta \Delta t) x_t + \sqrt{1 - (1 - \theta \Delta t)^2} \varepsilon_t,$$

which only contains the product $\theta \Delta t$ as a parameter. In this paper we thus set $\Delta t = 0.01$ without loss of generality. In the main text we also only consider OU noise as a replacement for strongly correlated Brownian motion and always set $\theta = 0.15$, as this is the most common default setting used in practice.[9] However, as noted in the discussion above, Ornstein-Uhlenbeck noise can also exhibit intermediate temporal correlation between white noise and Brownian motion, by setting $0 < \theta < 100$ (i.e. $0 < \theta \Delta t < 1$). This raises the question of whether there is a certain parameterization of OU noise which is as general as pink noise.

---

[9]We chose these values for $\Delta t$ and $\theta$ because these are the default choices the RL libraries we consider (Raffin et al., 2021; Pardo, 2020). Lillicrap et al. (2016) also recommend $\theta = 0.15$. If the variance is not corrected (we report these experiments in Sec. B), then the choice of $\Delta t$ does make a difference.

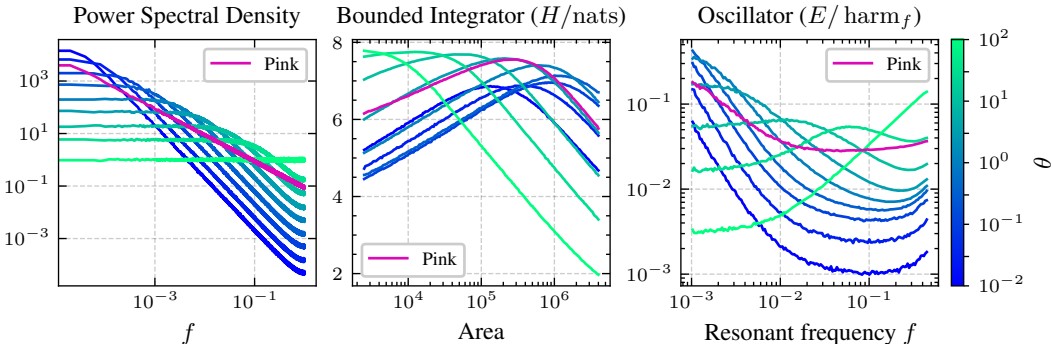

Figure A.4: Left: Power spectral densities of OU noise. OU noise interpolates between white noise and Brownian motion by changing the cutoff frequency of a low-pass filter which filters white noise. Center: Entropy achieved by OU noise of different $\theta$ on the bounded integrator environment. No $\theta$ achieves a higher worst-case entropy than pink noise. Right: Energy achieved by OU noise of different $\theta$ on the harmonic oscillator environment. No $\theta$ achieves a worst-case energy that comes close to the one of pink noise.

## A.2 GENERALITY OF ORNSTEIN-UHLENBECK NOISE

The way in which OU noise interpolates between white noise and Brownian motion by choosing $\theta \in (0, 100)$ is very different to colored noise with $\beta \in (0, 2)$. We have shown (e.g. in Fig. A.2) that colored noise with intermediate temporal correlation has a power-law power spectral density with intermediate exponent (or slope in the log-log plot). On the other hand, Ornstein-Uhlenbeck noise can be interpreted as a "leaky integration" of white noise, i.e. white noise passed through a low-pass filter. How "leaky" this integrator is, is controlled by the parameter $\theta$: if $\theta = 0$ then the integrator is ideal, resulting in integrated white noise (Brownian motion with diverging variance). If $\theta = 100$ (with $\Delta t = 0.01$), then the integrator is "completely leaky" (an all-pass filter) and the white noise passes through without being integrated. In terms of the power spectral density this change in $\theta$ corresponds to shifting the cutoff frequency of the low-pass filter. This is shown on the left in Fig. A.4 for $\theta \in \{0.01, 0.03, 0.1, 0.3, 1, 3, 10, 30, 100\}$.

For an action noise type to be *general* (cf. Sec. 6), we want it to work well on all environments. In the power spectral density plots, it can already be seen that pink noise distributes power over the frequencies much more "generally" than Ornstein-Uhlenbeck noise of any $\theta$: At any given frequency $f$, pink noise exhibits higher power than most values of $\theta$, and all values of $\theta$ have lower power than pink noise at most frequencies. Why this makes pink noise a more general action noise can be made more concrete by revisiting the bounded integrator and harmonic oscillator environments introduced in Sec. 6. The *generality* of a noise measures how robust it is to the choice or parameterization of the environment: The most general noise type is the one which performs best on the most adversarial environment parameterization. Thus, the most general $\theta$ for an environment parameterized by a parameter $\alpha$ solves the following optimization problem:

$$\max_{\theta} \min_{\alpha} \operatorname{perf}(\alpha, \theta),$$

where the performance metric $\operatorname{perf}(\alpha, \theta)$ should be normalized appropriately such that the maximum performance attainable for different values of $\alpha$ is identical. This can be ensured by simply dividing by the performance attained by the best $\theta$ for each value of $\alpha$:

$$\max_{\theta} \underbrace{\min_{\alpha, \theta'} \frac{\operatorname{perf}(\alpha, \theta)}{\operatorname{perf}(\alpha, \theta')}}_{\text{generality}(\theta)}. \tag{8}$$

This gives the worst-case performance of the most general noise in terms of the best possible performance achievable by changing the noise type on this worst-case environment. By replacing the expression $\operatorname{perf}(\alpha, \theta)$ by $\operatorname{perf}(\alpha, \text{pink})$ and removing the maximization over $\theta$, we can also calculate the generality of pink noise.

As discussed in Sec. 6, the performance of a noise type on the bounded integrator and oscillator environments is given by the achieved entropy and energy, respectively. This is shown for all values $\theta$ (as well as for pink noise) in Fig. A.4, where the parameterization parameter $\alpha$ is the environment size for the bounded integrator and the resonant frequency for the harmonic oscillator. It can already be seen that on both environments, for each choice of $\theta$ there exists a parameter $\alpha$ where the performance of $\theta$ is worse than the worst-case performance of pink noise. This can be quantified by calculating the generality of each $\theta$ and pink noise on these environments according to Eq. (8). On the bounded integrator, the maximum generality of OU noise is 77%, and on the oscillator environment the maximum generality is 9.1%. On both environments, the maximum is attained by $\theta = 3$. Pink noise achieves generalities of 79% and 22% on the bounded integrator and oscillator environments, respectively. This gives further evidence that pink noise is a good default.

## B    ADDITIONAL RESULTS

### B.1    TD3

In addition to MPO and SAC, we also performed all experiments from the main text on TD3. MPO and SAC parameterize a stochastic policy, meaning they *learn* the action noise scale as a function $\sigma(s)$ of the state. TD3, on the other hand, uses a deterministic policy, and the action noise is added independently of the state. Usually, the noise scale $\sigma$ is kept fixed over the course of training, and this how we handle it in our experiments as well. However, $\sigma$ is an important hyperparameter, and there is no single value that works well on all environments. Thus, we repeat our experiments with all of the values $\sigma \in \{0.05, 0.1, 0.3, 0.5, 1\}$, and 10 different random seeds.

In Fig. B.1, the results of the TD3 experiments with constant noise type are shown in the form of bootstrap distributions for the expected average performance, and compared to the same experiments on MPO and SAC, as well as to a Fig. 3-like plot where the influence of the agent has been normalized out. As we have an additional hyperparameter ($\sigma$), we first average the TD3 performance over all $\sigma$ values, before computing the average performance across tasks. The beneficial effect of pink noise

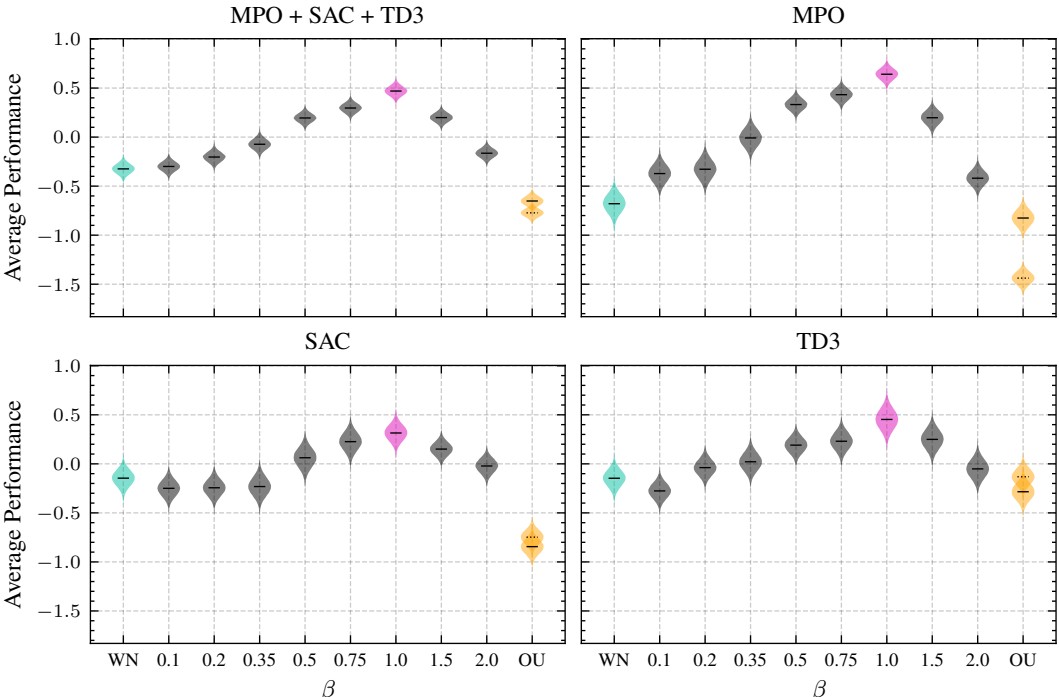

Figure B.1: All three algorithms (MPO, SAC, TD3) show a clear preference for pink action noise as measured by the average performance over the environments of Fig. 2. The results of the OU experiments with the uncorrected noise scale of $\sigma = 1$ are marked with a dotted median.

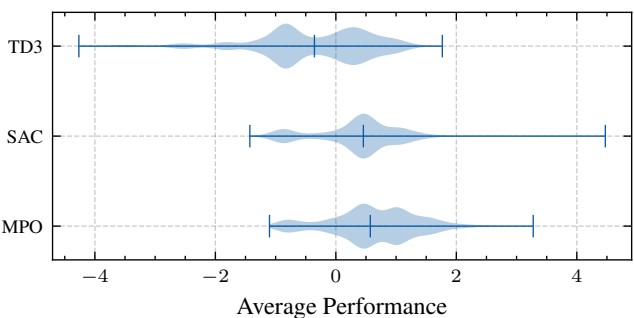

Figure B.2: Average performances across environments are combined from all $\beta$ values (incl. WN). It can be seen that TD3 is consistently outperformed by both MPO and SAC. A closer look at the mean performance over all $\beta$ values on each individual environment reveals that TD3 is outperformed on *all* environments by *both* MPO and SAC.

can be clearly seen on TD3 as well. In this figure we also show the results of Ornstein-Uhlenbeck noise with a noise scale of $\sigma = 1$ rather than the corrected noise scale of Eq. (7). Incidentally, these results also confirm Fujimoto et al. (2018)'s finding that, on TD3, white noise and OU noise (with $\theta = 0.15$) perform similarly.

The reason why we did not include TD3 into the analysis of the main text, is that we found TD3 to be consistently outperformed by both MPO and SAC. In Fig. B.2, the average performances across environments are combined from all $\beta$ values (incl. white noise), and shown for MPO, SAC and TD3. It can be seen that TD3 generally performs much worse than MPO and SAC. Looking at the mean performance over all $\beta$ values on each individual environment, TD3 is outperformed on *all* environments by *both* MPO and SAC. We thus decided to exclude TD3 from our main analysis.

## B.2   MPO & SAC

In the majority of this work, we measure performance in terms of the mean evaluation return over a training process. We use this method, because it implicitly measures both the final policy performance, and the sample efficiency (how quickly does the algorithm reach high performance). Most of the data we present is additionally normalized, which is necessary to aggregate performances over different environments, and thus it is often not very clear how exactly to interpret the results (other than recognizing statistical significance). In this section, we want to present some of our results in more familiar terms, namely learning curves and final policy performance.

To validate the approach of using the (mean) performance instead of the performance of the final policy, we have reproduced the results in Fig. 3 using the final policy performance (mean evaluation return in the last 5% of the training process), shown in Fig. B.3. In Fig. B.4, we show learning curves of white noise, pink noise, and OU noise on all environments for MPO and SAC. Both visualizations confirm our takeaway that pink noise is a better default action noise than white noise or OU noise. More detailed results can be found in Sec. H.

The bootstrap distributions for the expected average performance (such as in Figures 3, 4, B.1, and B.3) are constructed by randomly choosing one seed for each environment, yielding one scalar

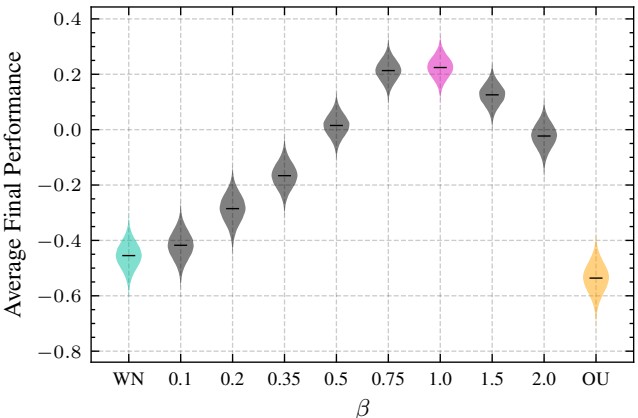

Figure B.3: The average *final* performance is like the average performance (see Sec. 4), but only uses the evaluation returns of the last 5% of training, thereby measuring the quality of the final learned policy. This figure shows the same analysis on MPO and SAC as Fig. 3, and demonstrates that pink noise is preferable also in terms of final policy performance.

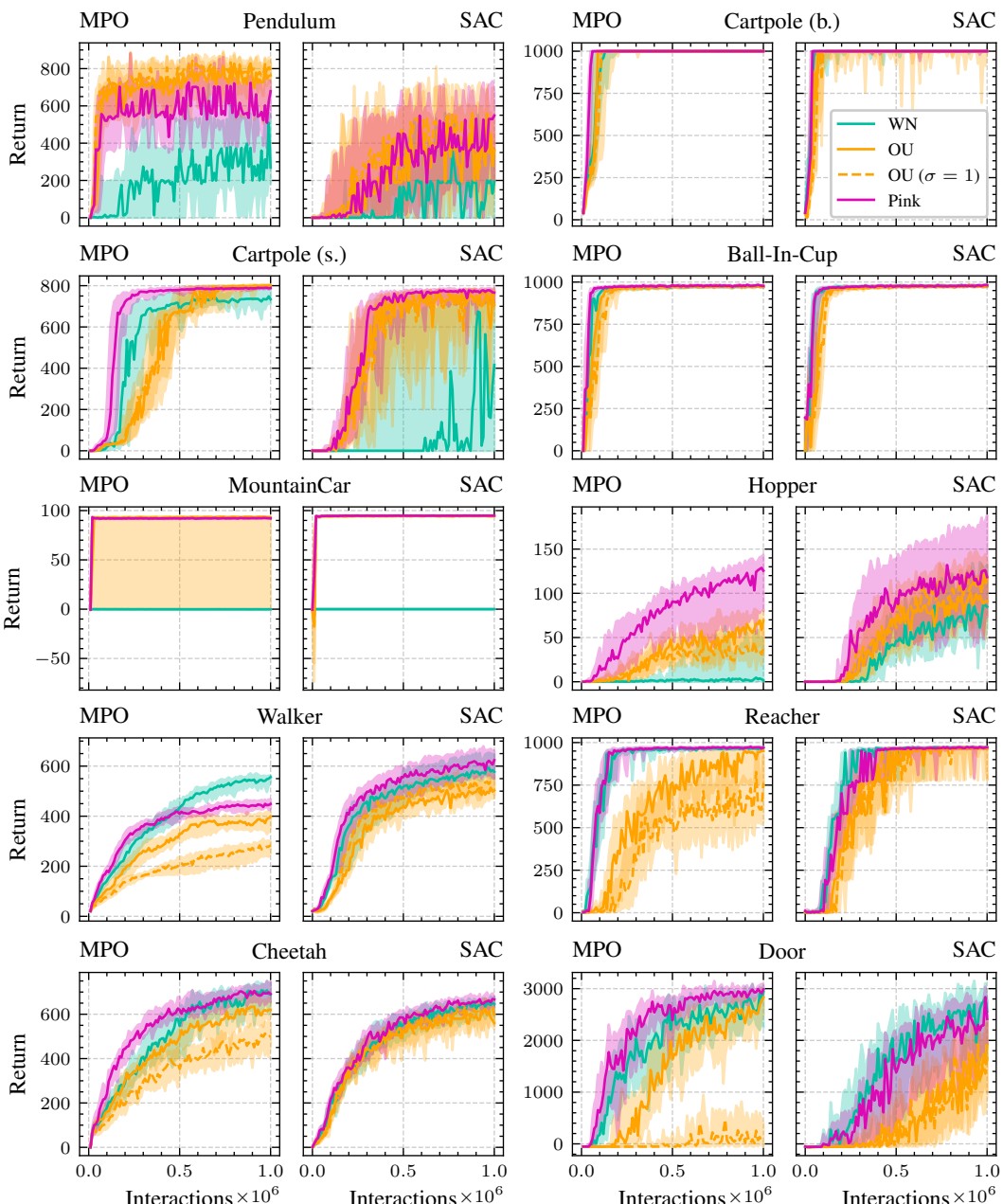

Figure B.4: Learning curves (median and interquartile range of evaluation returns) of the two baseline action noise types white noise (WN) and Ornstein-Uhlenbeck (OU) noise, as well as our suggestion of pink noise. It can be seen that pink noise, while not being better than both on all environments, is the best default choice. It is never outperformed by both white noise and OU noise, and routinely outperforms white noise (e.g. MountainCar), OU noise (e.g. Door), or both (e.g. Hopper).

(normalized) performance per environment, assuming all other variables like algorithm and noise type are fixed. Averaging these normalized performances (the reason that performances are normalized on each environment is so that this averaging is reasonable) gives an estimate for the *average performance* across environments of the given variables (e.g. noise type and algorithm). As there are $S$ different random seeds (typically $S = 20$), we can repeat this procedure $S$ times (with resampling) and take the mean of all $S$ average performance estimates, giving us an estimate for the expected average performance of the given variables. Doing this $N$ times (we use $N = 10^5$), the $N$ estimates for the expected average performance can be collected into a bootstrap distribution, as shown in these figures.

## C  Environments & Algorithms

We evaluate our method on 10 different tasks (see Fig. 2). Most of these are from the DeepMind Control Suite (DMC, Tassa et al., 2018), but we also use OpenAI Gym (Brockman et al., 2016) and the Adroit hand suite (Rajeswaran et al., 2018). The respective sources and exact IDs of all environments are compiled in Table C.1. See Sec. G for results on additional tasks.

| Environment | Source | ID |
|---|---|---|
| Pendulum | DMC | `pendulum (swingup)` |
| Cartpole (b.) | DMC | `cartpole (balance_sparse)` |
| Cartpole (s.) | DMC | `cartpole (swingup_sparse)` |
| Ball-In-Cup | DMC | `ball_in_cup (catch)` |
| MountainCar | Gym | `MountainCarContinuous-v0` |
| Hopper | DMC | `hopper (hop)` |
| Walker | DMC | `walker (run)` |
| Reacher | DMC | `reacher (hard)` |
| Cheetah | DMC | `cheetah (run)` |
| Door | Adroit | `door-v0` |

Table C.1: Environments used in this work (see also Fig. 2).

For our experiments, we relied on the TD3 and SAC implementations in Stable-Baselines3 (Raffin et al., 2021), as well as the MPO implementation in the Tonic RL library (Pardo, 2020). We only used the default hyperparameters of these algorithms, as provided by the libraries. Our own code for using colored noise with these libraries is made available online at `https://github.com/martius-lab/pink-noise-rl`.

## D  Bandit method details

To use a bandit algorithm to select the action noise color $\beta$ for a rollout, it is necessary to define the bandit reward, which should score a rollout in terms of the $\beta$ that was chosen. In our case, we use the rollout return (sum of rewards) as the score, as explained in Sec. 5.2. Additionally, we have to select a list of colors ("bandit arms") to search over: $B = (\beta_1, \beta_2, \ldots, \beta_K)$ (with $\beta_k \in [0, 2], \forall k$ in our case). If we assume that the bandit rewards (= rollout scores) are Gaussian distributed with a known standard deviation $\sigma$, we can use Bayesian inference to estimate the means ($\boldsymbol{\mu} \in \mathbb{R}^K$) of the reward distributions. A simple bandit algorithm we can use in this context is Thompson sampling, shown in Algorithm D.1 ($\mathbb{S}_+^K$ denotes the set of positive semi-definite $K \times K$ matrices). The relationships between the random variables are shown in the Bayesian network in Figure D.1a.

---

**Algorithm D.1:** Thompson Sampling

**Input:** Arms $B = (\beta_1, \ldots, \beta_K)$,
           Reward distributions std $\sigma$
Initialize $\boldsymbol{m} \in \mathbb{R}^K, \Sigma \in \mathbb{S}_+^K$
**for** $i \in \mathbb{N}$ **do**
     Sample $\boldsymbol{q} \sim \mathcal{N}(\boldsymbol{m}, \Sigma)$
     $a_i \leftarrow \arg\max_{k \in \{1, \ldots, K\}} q_k$
     $\tau_i \leftarrow$ Run rollout with $\beta_{a_i}$
     $r_i \leftarrow$ score of rollout $\tau_i$
     Do Bayesian update of $\boldsymbol{m}, \Sigma$ using
     $\{a_j, r_j\}_{j=1}^i, \sigma$
**end**

---

There is a second strong assumption in the Thompson sampling algorithm shown in Algorithm D.1 (similarly for other algorithms like UCB): it assumes that the reward distributions are stationary, i.e. that they don't change over time. This is not the case in the context of reinforcement learning: if the rollout score $r_i$ is defined as the return, then, if the reinforcement learning algorithm works, it should naturally be the case that the policy improves over time, and thus, on average, $r_i > r_j$ for $i \gg j$. This setting of non-stationary bandit distributions can be addressed by using a sliding-window approach (e.g. Garivier & Moulines, 2008): instead of updating the belief parameters $\boldsymbol{m}, \Sigma$ with respect to the whole history of observations, only keep a window of the last $N$ rollouts.

There remains one other problem: how do we choose the prior parameters $\boldsymbol{m}$ and $\Sigma$ and the variance $\sigma^2$ of the reward distributions? For $\Sigma$, the easiest solution is to assume independent arms, i.e. make

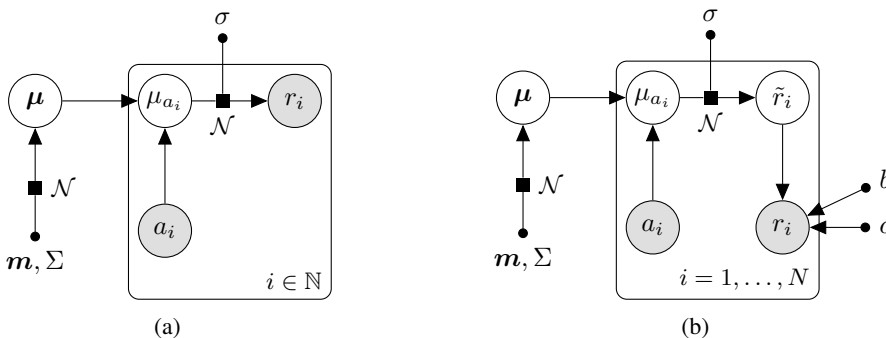

Figure D.1: (a) A Bayesian bandit with Gaussian reward distributions. The rewards from arm $k$ are sampled from $\mathcal{N}(\mu_k, \sigma)$. Thompson sampling (Alg. D.1) can infer $\boldsymbol{\mu}$ while trading off exploration and exploitation. (b) By introducing the constants $b$ and $c$, the algorithm can be made scale invariant by performing Thompson sampling with respect to the normalized reward $\tilde{r}_i = (r_i - b)/c$.

$\Sigma$ diagonal. This is not necessarily the most efficient solution, as one can imagine that two similar $\beta$ values will also perform similarly in their rollouts.[10] For $\boldsymbol{m}$, the non-stationarity becomes a problem: again assuming we use the rollout return as a score, these scores will probably be much lower at the beginning of training than at the end. Additionally, we might not even know the scale of returns in a task. To account for this, it would be necessary to make the prior variances $\Sigma_{kk}$ very large/uninformed. Similarly, $\sigma$ needs to be large, to account for the unknown scale of the bandit reward spread. However, this would mean that many more samples (rollouts) are necessary to tighten the belief distributions. This is a problem, especially because we only have a small set of $N$ rollouts when using the sliding-window method.

The ideal would be a bandit method which is invariant with respect to affine transformations of the rewards, in the sense that it would make no difference if all rewards $r$ were transformed to be $br + c$ for some constants $b > 0$ and $c \in \mathbb{R}$ for all arms. In Fig. D.1b, this situation is shown in a Bayesian network. Here, the generative process is almost the same as before (see Fig. D.1a), except that the reward $\tilde{r}_i$ is scaled and translated by $r_i = b\tilde{r}_i + c$ before observation. If, as shown, the constants $b$ and $c$ are independent of the chosen arm and stay constant within the window, it is possible to optimize them via maximum marginal likelihood, given the window of past observations of $r_i$.

The bandit inference task is to infer the distributional means $\boldsymbol{\mu} = (\mu_1, \ldots, \mu_k)$ from the actions (color indices) $\boldsymbol{a} = (a_i)_{i=1}^N$ and rewards (rollout scores) $\boldsymbol{r} = (r_i)_{i=1}^N$. We set the prior means of the belief distributions to 0 ($\boldsymbol{m} = \boldsymbol{0}$), because we want the normalized reward distributions to be centered around 0. For now, we don't fix $\Sigma$, but let it be any positive semi-definite $K \times K$ matrix. The generative model for $\boldsymbol{r}$ is defined via the following prior and likelihood function:

$$p(\boldsymbol{\mu} \mid \Sigma) = \mathcal{N}(\boldsymbol{\mu} \mid \boldsymbol{0}, \Sigma) \tag{9}$$

$$p(\boldsymbol{r} \mid \boldsymbol{\mu}, \boldsymbol{a}, b, c, \sigma) = \prod_i \mathcal{N}(r_i \mid b\mu_{a_i} + c, (b\sigma)^2) \tag{10}$$

These lead us to the following evidence/marginal likelihood function:

$$p(\boldsymbol{r} \mid \boldsymbol{a}, b, c, \sigma, \Sigma) = \prod_i p(r_i \mid a_i, b, c, \sigma, \Sigma) \tag{11}$$

$$= \prod_i \int p(r_i \mid \boldsymbol{\mu}, a_i, b, c, \sigma) \, p(\boldsymbol{\mu} \mid \Sigma) \, \mathrm{d}\boldsymbol{\mu} \tag{12}$$

$$= \prod_i \int \mathcal{N}(r_i \mid b\mathbf{e}_{a_i}^\top \boldsymbol{\mu} + c, (b\sigma)^2) \, \mathcal{N}(\boldsymbol{\mu} \mid \boldsymbol{0}, \Sigma) \, \mathrm{d}\boldsymbol{\mu} \tag{13}$$

$$= \prod_i \mathcal{N}(r_i \mid b\mathbf{e}_{a_i}^\top \boldsymbol{0} + c, (b\sigma)^2 + b\mathbf{e}_{a_i}^\top \Sigma b\mathbf{e}_{a_i}) \tag{14}$$

---

[10]We also tried a different approach by using a modified RBF kernel matrix to account for covariance between the arms, but the results were essentially the same as with independent arms.

$$= \prod_i \mathcal{N}(r_i \mid c, b^2(\sigma^2 + \Sigma_{a_i a_i})), \tag{15}$$

where we used canonical basis vectors to represent $\mu_{a_i} = \mathbf{e}_{a_i}^\top \boldsymbol{\mu}$. For maximization, it is convenient to work with the log-evidence:

$$\log p(\boldsymbol{r} \mid \boldsymbol{a}, b, c, \sigma, \Sigma) = \log \prod_i \mathcal{N}(r_i \mid c, b^2(\sigma^2 + \Sigma_{a_i a_i})) \tag{16}$$

$$= \sum_i -\frac{1}{2} \log\bigl(2\pi b^2(\sigma^2 + \Sigma_{a_i a_i})\bigr) - \frac{(c - r_i)^2}{2b^2(\sigma^2 + \Sigma_{a_i a_i})} \tag{17}$$

$$=: L(b, c) \tag{18}$$

We can now maximize the evidence by setting the partial derivatives to $0$:

$$\partial_c L(b, c) \propto \sum_i (c - r_i) = 0 \tag{19}$$

$$\partial_b L(b, c) = \sum_i \frac{-1}{b} + \frac{(c - r_i)^2}{b^3(\sigma^2 + \Sigma_{a_i a_i})} = 0 \tag{20}$$

Solving these equations gives us

$$c = \frac{1}{N} \sum_i r_i \tag{21}$$

$$b^2 = \frac{1}{N} \sum_i \frac{(c - r_i)^2}{\sigma^2 + \Sigma_{a_i a_i}}. \tag{22}$$

Using these values, we can "reconstruct" the unscaled/normalized reward

$$\tilde{r}_i = \frac{r_i - c}{b} \tag{23}$$

and perform Thompson sampling with respect to $\tilde{r}_i$. This *normalized Thompson sampling* algorithm, including the sliding window modification, is presented in Algorithm D.2.

---

**Algorithm D.2:** Normalized TS

**Input:** Arms $B = (\beta_1, \ldots, \beta_K)$,
  Window size $N$
Initialize
  $\boldsymbol{m} \leftarrow \mathbf{0} \in \mathbb{R}^K, \Sigma \in \mathbb{S}_+^K, \sigma \leftarrow 1$
**for** $l \in \mathbb{N}$ **do**
  $i \leftarrow l \mod N$
  $M \leftarrow \min\{l, N\}$
  Sample $\boldsymbol{q} \sim \mathcal{N}(\boldsymbol{m}, \Sigma)$
  $a_i \leftarrow \arg\max_{k \in \{1, \ldots, K\}} q_k$
  $\tau_i \leftarrow$ Run rollout with $\beta_{a_i}$
  $r_i \leftarrow$ score of rollout $\tau_i$
  $c \leftarrow \frac{1}{M} \sum_{j=1}^M r_j$
  $b \leftarrow \sqrt{\frac{1}{M} \sum_{j=1}^M \frac{(c - r_j)^2}{\sigma^2 + \Sigma_{a_j a_j}}}$
  $\tilde{r}_i \leftarrow \frac{r_i - c}{b}$
  Do Bayesian update of $\boldsymbol{m}, \Sigma$ using
    $\{a_j, \tilde{r}_j\}_{j=1}^M, \sigma$
**end**

---

Next, we want to show that this method is indeed invariant to affine transformations of the bandit reward.

**Proposition 1.** *The posterior distribution over $\boldsymbol{\mu}$ in the normalized bandit algorithm (Alg. D.2) is identical for the observations $\boldsymbol{r} = (r_1, \ldots, r_N)$ and $\boldsymbol{r}' = b'\boldsymbol{r} + c'$, for all $b' > 0$ and $c' \in \mathbb{R}$. In other words, the algorithm is invariant to a scaling and translation of the rewards.*

*Proof.* In this setting, the observed rewards $r_i$ are normalized to

$$\tilde{r}_i = \frac{r_i - c(\boldsymbol{r})}{b(\boldsymbol{r})} \tag{24}$$

with

$$c(\boldsymbol{r}) = \frac{1}{N} \sum_{i=1}^N r_i \tag{25}$$

$$b(\boldsymbol{r}) = \sqrt{\frac{1}{N} \sum_{i=1}^N \frac{(c(\boldsymbol{r}) - r_i)^2}{\sigma^2 + \Sigma_{a_i a_i}}}. \tag{26}$$

To prove the invariance of the algorithm, we will simply show that this normalized reward is the same for both sets of observations, i.e. that $\tilde{\boldsymbol{r}} = \tilde{\boldsymbol{r}}'$. Then, clearly, the posteriors $p(\boldsymbol{\mu} \mid \tilde{\boldsymbol{r}})$ and $p(\boldsymbol{\mu} \mid \tilde{\boldsymbol{r}}')$ will also be the same. Expanding $\tilde{\boldsymbol{r}}'$, we get:

$$\tilde{\boldsymbol{r}}' = \frac{\boldsymbol{r}' - c(\boldsymbol{r}')}{b(\boldsymbol{r}')} \tag{27}$$

$$= \frac{b'\boldsymbol{r} + c' - c(b'\boldsymbol{r} + c')}{b(b'\boldsymbol{r} + c')} \tag{28}$$

$$= \frac{b'\boldsymbol{r} + c' - \frac{1}{N}\sum_{i=1}^{N}(b'r_i + c')}{\sqrt{\frac{1}{N}\sum_{i=1}^{N}\frac{(\frac{1}{N}\sum_{j=1}^{N}(b'r_j + c') - (b'r_i + c'))^2}{\sigma^2 + \Sigma_{a_i a_i}}}} \tag{29}$$

$$= \frac{b'\boldsymbol{r} + c' - b'\frac{1}{N}\sum_{i=1}^{N}r_i - c'}{\sqrt{\frac{1}{N}\sum_{i=1}^{N}\frac{(b'\frac{1}{N}\sum_{j=1}^{N}r_j + c' - b'r_i - c')^2}{\sigma^2 + \Sigma_{a_i a_i}}}} \tag{30}$$

$$= \frac{b'(\boldsymbol{r} - c(\boldsymbol{r}))}{\sqrt{\frac{1}{N}\sum_{i=1}^{N}\frac{b'^2(c(\boldsymbol{r}) - r_i)^2}{\sigma^2 + \Sigma_{a_i a_i}}}} \tag{31}$$

$$= \frac{\boldsymbol{r} - c(\boldsymbol{r})}{b(\boldsymbol{r})} \tag{32}$$

$$= \tilde{\boldsymbol{r}} \tag{33}$$

Thus, we can conclude that the reward normalization indeed guarantees invariance to affine reward transformations in algorithms such as Thompson sampling. $\qquad\square$

With this reward normalization, the prior parameters $m$ (of $\boldsymbol{m} = m\mathbf{1}$) and $s$ (of $\Sigma = s^2 I$) become redundant. We have already set $\boldsymbol{m} = \mathbf{0}$, and we now also set the prior variances $\Sigma_{kk}$ to 1. This encourages the algorithm to keep the normalized mean estimates $\mu_k$ approximately $\mathcal{N}(0, 1)$-distributed. The "likelihood" parameter $\sigma$ remains to be tuned, but it is now not necessary to account for the large uncertainty in the reward scale, as $\sigma$ is only concerned with the normalized reward. In our experiments we always set $\sigma = 1$.

### D.1 BANDIT VS. RANDOM

Although we found the normalized bandit algorithm (Alg. D.2) to work well on simple non-stationary tasks, in the RL setting (for choosing $\beta$) the performance was just as that of a random $\beta$ selection for every rollout. In Table D.1, we list the results of a Welch $t$-test, testing for inequality of the performance distributions achieved by the bandit algorithm and random $\beta$ selection on every environment. It can be seen that the two methods are statistically indistinguishable. This shows that the bandit method does not work as intended, as "random arm selection" should be an easy

| Environment | Bandit $\neq$ Random | $p$ |
|---|---|---|
| Pendulum | ✗ | 0.98 |
| Cartpole (b.) | ✗ | 0.09 |
| Cartpole (s.) | ✗ | 0.67 |
| Ball-In-Cup | ✗ | 0.87 |
| MountainCar | ✗ | 0.54 |
| Hopper | ✗ | 0.09 |
| Walker | ✗ | 0.15 |
| Reacher | ✗ | 0.70 |
| Cheetah | ✗ | 0.20 |
| Door | ✗ | 0.59 |

Table D.1: Bandit vs. Random (Welch $t$-test)

baseline to outperform. The reason for this is probably due to the rollout return not being informative enough as a bandit reward signal.

# E    SOLVING MOUNTAINCAR BY FFT

MountainCar is a very simple environment. Although its dynamics are almost those of a harmonic oscillator, there is a difference to the oscillator environment from Sec. 6: MountainCar's oscillation dynamics are non-linear. At the bottom of MountainCar's valley (see Fig. 2), the small-angle approximation of a non-linear oscillator may be used, but for the motion to go up to the top, the behavior is different from simple harmonic motion. Nevertheless, we can use this insight to develop a very simple open-loop control algorithm to solve this environment, by running one rollout without applying any action (just letting the mountain make the car go back and forth a bit), then analyzing the resulting trajectory and inferring the hill's (small-angle) resonant frequency (via the Fast Fourier Transform algorithm). Finally, we can control the car by simply swinging it back and forth at the resonant frequency. This algorithm, which works very well on this task, is shown below.

```python
import gym
import numpy as np
from scipy.fft import rfft

# Initialize environment
env = gym.make('MountainCarContinuous-v0')
T = env._max_episode_steps

# Run a single rollout with no force. Save x-coordinate to `x`.
obs = env.reset()
x = [obs[0]]
for t in range(T):
    obs, *_ = env.step([0])
    x.append(obs[0])

# Find resonant frequency = highest peak of FFT (excluding DC)
f = (np.argmax(abs(rfft(x))[1:]) + 1) / (T + 1)

# Action plan (harmonic excitation)
a = np.sin(2*np.pi*f * np.arange(T))

# Test on 1000 rollouts
N = 1000
solved = 0
for i in range(N):
    env.reset()
    for t in range(T):
        _, r, _, _ = env.step([a[t]])
        if r > 0:
            solved += 1
            break

print(f"Solved: {solved/N * 100:.0f}%.")  # prints "Solved: 100%."
```

# F    TOY ENVIRONMENT DETAILS

## F.1    OSCILLATOR ENVIRONMENT

The oscillator environment of Sec. 6, which we make available online as a gym environment (https://github.com/onnoeberhard/oscillator-gym), models the 1-dimensional motion of a particle of mass $m$, attached to the origin by an ideal spring of stiffness $k$, damped with friction coefficient $b$, and driven by a force (the action) $F$. This motion is described by the ordinary differential equation

$$m\ddot{x} = F - b\dot{x} - kx, \tag{34}$$

where $x$ is the particle's position. In our experiments we set the friction coefficient $b$ to zero, i.e. the system is undamped. This setup is then called a *simple harmonic oscillator*. The energy of the

oscillator is the sum of kinetic and potential energy:

$$E = \frac{1}{2}m\dot{x}^2 + \frac{1}{2}kx^2. \tag{35}$$

The resonant frequency is:

$$f = \frac{1}{2\pi}\sqrt{\frac{k}{m}}. \tag{36}$$

As we want to configure the oscillator to have a given resonant frequency $f$, we need to find $m$ and $k$ accordingly. To get a unique solution, we impose a second constraint: the energy at $x = 1$ and $\dot{x} = 0$ should be $E = 2\pi^2$. If we now solve the two equations (35) and (36) for $m$ and $k$, imposing the constraint on $E$, we get the solution

$$k = 4\pi^2 \tag{37}$$

$$m = \frac{1}{f^2} \tag{38}$$

to set the resonant frequency. In Fig. F.1, a few pure-noise trajectories (akin to Fig. 1) are shown on the oscillator environment.

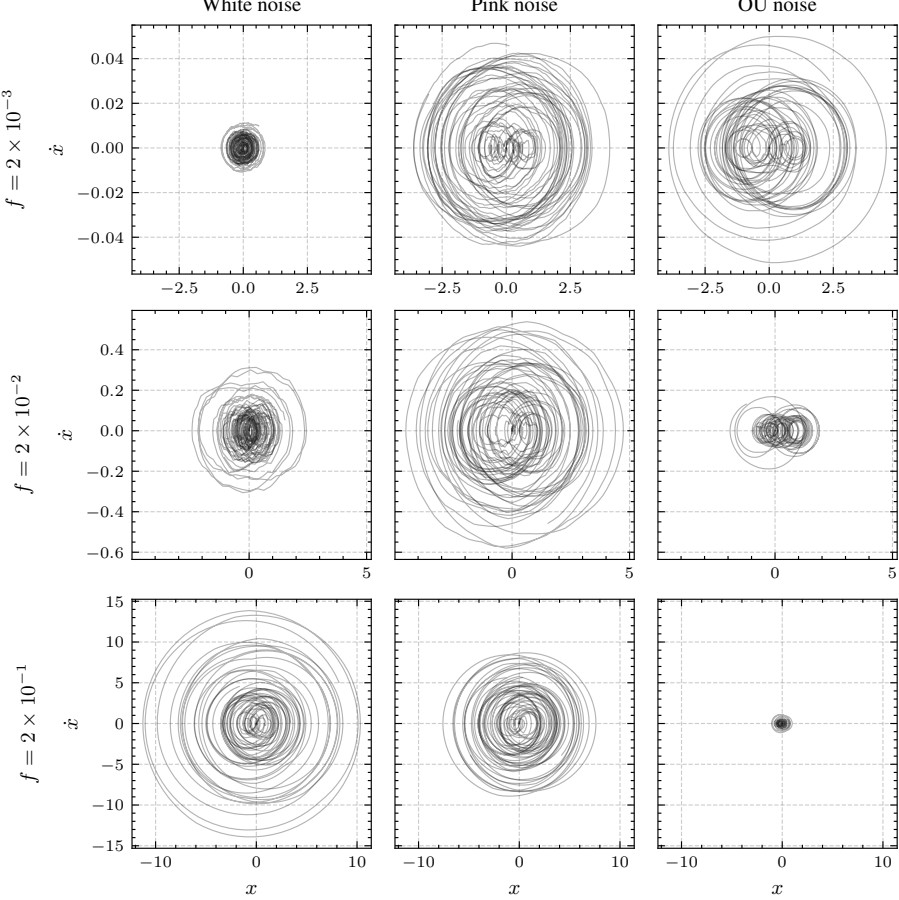

Figure F.1: Trajectories on the oscillator environment. For each of the 3 resonance frequencies $f \in \{0.002, 0.02, 0.2\}$, we sample 5 action noise signals of length $\frac{10}{f}$ of white noise, pink noise and OU noise. We can see what was already shown in Fig. 5: pink noise is much less sensitive to the parameterization than white noise and OU noise, and always manages to excite the oscillator up to a certain amplitude. White noise and OU noise only work well in the high- and low-frequency regime, respectively.

### F.2 BOUNDED INTEGRATOR ENVIRONMENT

The bounded integrator environment of Sec. 6 has very simple dynamics:

$$s_{t+1} = \text{clip}(s_t + a_t, -c, c), \tag{39}$$

where $s_0 = 0$ and $c$ is the parameter determining the size of the environment. Thus, the "area" in Fig. 5 is given by $c^2$. In Fig. 1, this parameter is fixed at $c = 250$, and in Fig. F.2 these trajectories (center row) are compared to trajectories on a smaller (top row) and a larger environment (bottom row), in a similar spirit to Fig. F.1.

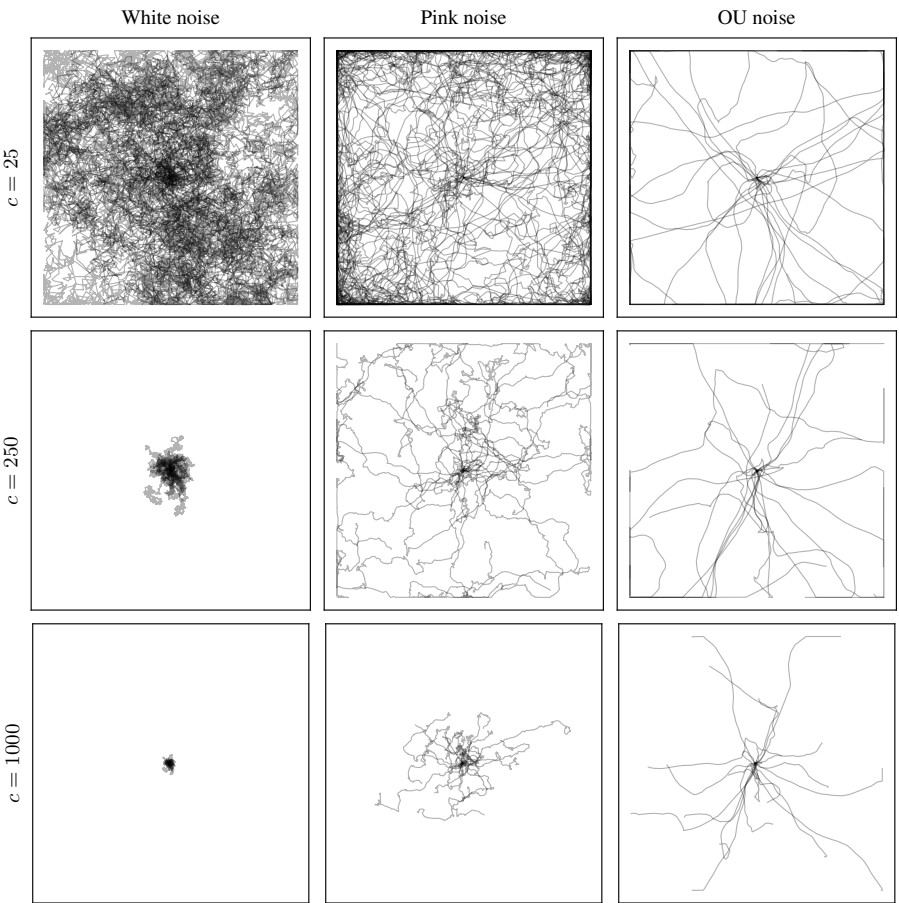

Figure F.2: Trajectories on the bounded integrator environment. For each of the 3 environment sizes $c \in \{25, 250, 1000\}$, we sample 20 action noise signals of length 1000 steps of white noise, pink noise, and OU noise. We can see what was already shown in Fig. 5: pink noise is less sensitive to the parameterization than white noise (which is too slow to explore the medium and large environments) and OU noise (which, on the medium and small environments, gets stuck at the edges and fails to explore the interior).

## G   ADDITIONAL ENVIRONMENTS

In addition to the environments described in Sec. C, we also ran experiments on several tasks from the "MuJoCo-Maze" suite (`https://github.com/kngwyu/mujoco-maze`). The results are shown in Fig. G.1 in the form of learning curves and the average performance (cf. Sec. B.2) of each noise type over all six environments. We tested white noise, pink noise, and (variance-corrected, cf. Sec. A.1) Ornstein-Uhlenbeck noise, and trained MPO and SAC on all environments for $10^6$ interactions using 20 seeds. Pink noise again outperforms white noise and Ornstein-Uhlenbeck noise as a default choice across environments. These experiments were conducted to verify our method on

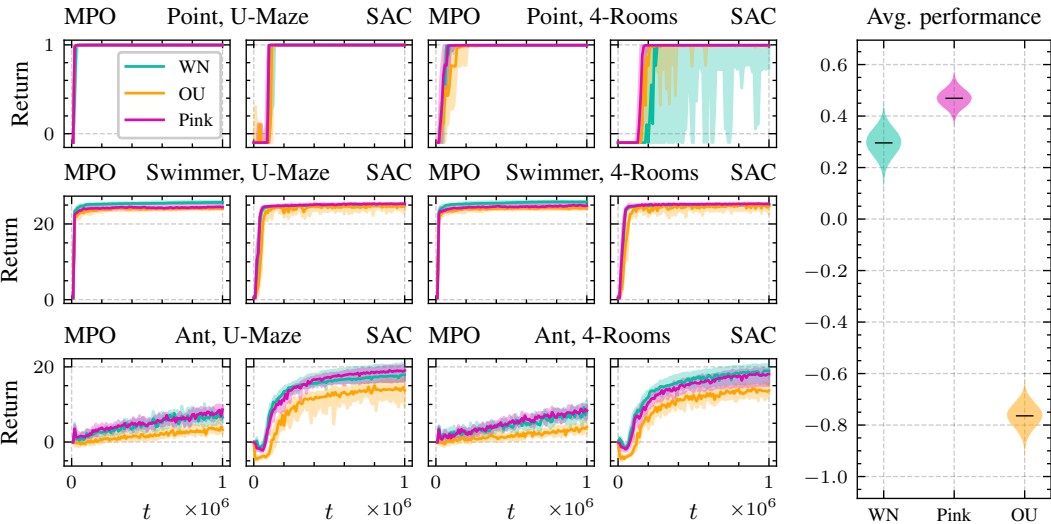

Figure G.1: Performances of white noise, pink noise, and OU noise on several "MuJoCo-Maze" tasks (all with sparse rewards). It can be seen that pink noise is the best default choice of the three, as it has the highest average performance (Sec. B.2).

a different set of problems and are thus not included in the main analysis. The results provide further evidence in support of our main takeaway: *pink noise makes a very good default action noise*.

## H DETAILED RESULTS

| Environment | Agent | Final Performance | | | Mean Performance | | | | | |
|---|---|---|---|---|---|---|---|---|---|---|
| | | WN | OU | Pink | WN | OU | Pink | Oracle | Anti | Gain |
| Pendulum | MPO | 311 | 702 | 574 | 247 | 651 | 558 | 670 | 239 | 430 |
| | SAC | 224 | 350 | 446 | 158 | 283 | 294 | 361 | 158 | 202 |
| Cartpole (b.) | MPO | 999 | 1000 | 1000 | 928 | 940 | 967 | 967 | 928 | 39 |
| | SAC | 960 | 908 | 958 | 939 | 890 | 941 | 950 | 890 | 59 |
| Cartpole (s.) | MPO | 703 | 784 | 788 | 535 | 499 | 666 | 666 | 489 | 177 |
| | SAC | 377 | 608 | 730 | 226 | 459 | 532 | 533 | 159 | 374 |
| Ball-In-Cup | MPO | 974 | 973 | 978 | 926 | 909 | 948 | 948 | 909 | 39 |
| | SAC | 976 | 975 | 979 | 930 | 901 | 933 | 941 | 901 | 39 |
| MountainCar | MPO | 13 | 56 | 92 | 13 | 52 | 91 | 92 | 13 | 78 |
| | SAC | 0 | 90 | 94 | 0 | 89 | 93 | 93 | 0 | 93 |
| Hopper | MPO | 25 | 62 | 108 | 14 | 34 | 69 | 69 | 14 | 54 |
| | SAC | 89 | 94 | 119 | 43 | 53 | 77 | 80 | 43 | 36 |
| Walker | MPO | 530 | 377 | 448 | 384 | 284 | 363 | 390 | 284 | 106 |
| | SAC | 593 | 506 | 602 | 437 | 363 | 471 | 472 | 363 | 108 |
| Reacher | MPO | 956 | 856 | 966 | 864 | 600 | 871 | 888 | 581 | 306 |
| | SAC | 955 | 914 | 940 | 776 | 653 | 745 | 776 | 653 | 122 |
| Cheetah | MPO | 666 | 612 | 678 | 481 | 440 | 543 | 543 | 440 | 103 |
| | SAC | 631 | 577 | 640 | 469 | 439 | 483 | 502 | 439 | 63 |
| Door | MPO | 2586 | 2492 | 2909 | 1830 | 1376 | 2207 | 2207 | 1376 | 830 |
| | SAC | 2192 | 1535 | 2195 | 1332 | 546 | 1183 | 1332 | 546 | 785 |

Table H.1: Comparison of final policy performance (see Sec. B.2) and mean performance over the training process (Sec. 4) on all environments. Results are averaged across seeds, and shown for white noise (WN), Ornstein-Uhlenbeck noise (OU), and pink noise (Pink) as action noise on MPO and SAC. Additionally, the Oracle and Anti-Oracle ("Anti") performances are shown. The gain between these (rightmost column) represents the difference achievable by changing the noise type, and is the basis for the "performance gain" measure used in Sec. 4.2.

