# OpenReview forum: "Pink Noise Is All You Need: Colored Noise Exploration in Deep Reinforcement Learning"
_ICLR.cc/2023/Conference — ICLR 2023 notable top 25%_

### Official Review · Reviewer_EijA · 2022-10-18

**Confidence:** 5
**Correctness:** 3
**Technical Novelty And Significance:** 3
**Empirical Novelty And Significance:** 3
**Recommendation:** 8

**Clarity, Quality, Novelty And Reproducibility:**

**Clarity**

The paper is mostly clear and well-written.

**Novelty**

The idea of using temporally-correlated noise is not totally new, while pink noise has not been comprehensively investigated. I would say the novelty is moderate, thus its effectiveness should be more focused.

**Quality**

The idea is most heurestic and the results are mostly empirical. The paper will be impactful in RL if the authors can consolidate their results with my suggestions above.

**Reproducibility**

It should be easy to reproduce though I have not checked the code.



**Strength And Weaknesses:**

**[Strength]**
- The idea is simple, intuitive and agnostic to algorithm and task.
- Great experimental results on robotic control tasks
- The writing is easy to follow.

**[Liminations & Questions]**
- One major concern is about the choice of hyper-parameters for OU noise ($\theta$, $\Delta t$ and $\sigma$ in Equation 6). I have tried OU noise for motor exploration with myself in many tasks. The OU noise oftern outperforms white noise (My choice is $\theta$=0.3, $\Delta t$=1). And $\sigma$ is selected so that the standard deviation (STD) is approximately the same as the original Gaussian white noise with $\sigma = STD * sqrt(\theta) * sqrt(2)$. Since the paper searched $\beta$ for colored noise,  a search of $\theta$  and $\Delta t$ should also be conducted for OU noise.

- Also, I think a fair comparison is to keep the STD of white noise, colored noise and OU noise the same in experiments, did the authors do so?

- Another limitation of this work is that it is unclear how pink noise works on tasks rather than MuJoCo-like robotic control tasks. Since the method is simple, it should be straightforward to apply on more diverse benchmarks.

- Have the authors considered applying temporally-correlated noise on discrete action space?



**Summary Of The Paper:**

This paper suggested that colored noise for motor exploration in RL, in particular Pink noise ($\beta$ = 1) is a better choice than using Gaussian white noise and OU noise in many control tasks. The method is simple and versatile for online RL, and effective on the being tested environments. The main contribution is that we may improve online RL performance on most task with a very simple modification to the motor noise.



**Summary Of The Review:**

I think the paper touched an important yet not fully addressed question in RL -- the choice of motor noise. The proposed method could have the potential to be a common design choice in RL if the authors can show the effectiveness of pink noise using more diverse tasks. Overall, I slightly lean toward rejection, but there is a large space to improve.


-------------
### Update

Since the rebuttal has addressed my major concern about OU-noise, I increased my score and recommend acceptance.

---

> ### Author Response · Authors · 2022-11-17
> **Response to Reviewer EijA**
>
> Thank you very much for your insightful review of our paper. We think the changes we made as a response have made our work much more valuable.
>
> > One major concern is about the choice of hyper-parameters for OU noise ($\theta$, $\Delta t$ and $\sigma$ in Equation 6). I have tried OU noise for motor exploration with myself in many tasks. The OU noise oftern outperforms white noise (My choice is $\theta=0.3, \Delta t=1$). And $\sigma$ is selected so that the standard deviation ($\mathrm{STD}$) is approximately the same as the original Gaussian white noise with $\sigma = \mathrm{STD} \sqrt{2\theta}$. Since the paper searched $\beta$ for colored noise, a search of $\theta$ and $\Delta t$ should also be conducted for OU noise.
> > Also, I think a fair comparison is to keep the $\mathrm{STD}$ of white noise, colored noise and OU noise the same in experiments, did the authors do so?
>
> Thank you for raising these points. The white noise and colored noise we use are both marginally standard Gaussian, i.e. if $\varepsilon_{1:T}$ is the noise signal for an episode, then the marginal distribution $p(\varepsilon_t) = \mathcal N(0, 1)$, so in your notation $\mathrm{STD} = 1$. Indeed, by using Ornstein-Uhlenbeck noise with a noise scale of $\sigma = 1$, this is not the case. We thus reran all our OU noise experiments with the corrected noise scale $\sigma = \sqrt{\frac{1 - (1 - \theta\Delta t)^2}{\Delta t}}$, which ensures that the OU noise standard deviation is $\mathrm{STD} = 1$ (in the limit). Indeed, if $\Delta t = 1$ and $\theta \ll 1$, this expression effectively reduces to the correction of $\sqrt{2\theta}$ that you suggest. A full derivation of this result is included in Section A.1. The previous OU experiments with $\sigma = 1$ (which we still feel have value, as $\sigma = 1$ is a common choice) are now included in Section B. Overall, the performances achieved with the variance-corrected OU noise are very similar to the ones achieved with $\sigma = 1$. As the new experiments affect the performance normalization constants and the oracle/anti-oracle, many numbers in the text and tables change slightly, however all conclusions that are drawn from these numbers remain unchanged.
>
> With the variance correction, the parameters $\theta$ and $\Delta t$ can be collapsed into a single parameter $\theta\Delta t$ (see Section A.1 for a proof). A search over this parameter is, however, still too expensive within the two-week discussion phase, as it would result in running several thousand experiments. Instead of this, we have thus decided to analyze the behavior of OU noise with different parameterizations (including the suggested setting of $\theta = 0.3, \Delta t = 1$, which is equivalent to the setting we call $\theta = 30$ with $\Delta t = 0.01$) on the two toy environments we discuss in Section 6 (the bounded integrator and harmonic oscillator). In the new Section A.2, we use these environments to empirically demonstrate how pink noise is more general than Ornstein-Uhlenbeck noise in terms of worst-case performance: For each individual environment, there is a best choice for $\theta$, but no single $\theta$ is better under the worst environment parameterization than pink noise is in the worst case. This shows that for OU noise it is important to select the correct hyperparameter $\theta$, whereas pink noise, which has no hyperparameter, performs well without tuning.
>
> > Another limitation of this work is that it is unclear how pink noise works on tasks rather than MuJoCo-like robotic control tasks. Since the method is simple, it should be straightforward to apply on more diverse benchmarks
>
> We found the set of environments we chose to be representative of the continuous control domain. During the discussion phase we did not have enough time to run new experiments, but we are happy to run more experiments for the camera-ready version, and are open to environment suggestions.
>
> > Have the authors considered applying temporally-correlated noise on discrete action space?
>
> Due to the lack of a similarity measure between actions in discrete action spaces, the concept of correlation cannot be straightforwardly translated. The only applicable type of temporally correlated action signal is one with action repeats. This idea has been studied in other papers.
>
> We hope that we have addressed all your concerns and, if this is the case, that this can be reflected in the rating.

---

> > ### Comment · Reviewer_EijA · 2022-11-19
> > **Thanks for the response**
> >
> > Thanks for the response. I appreciate the revisions about OU-noise. I increased my score to 8, assuming that the authors will show more results on different task sets.  The follows are some additional comments
> >
> > >  we are happy to run more experiments for the camera-ready version, and are open to environment suggestions.
> >
> > Even in the case that the performance of pink noise is unsatisfying, the RL community would benefit from the empirically results and discussions reflecting when and why colored noise is helpful or not. I thought temporally-correlated noise is beneficial for exploration when the rewarded state is hard to reach with white noise, e.g., in mountain car. While the existing results are based on non-sparse-reward tasks (MuJoCo robotic control), how about trying pink noise on some continuous control environment with sparse rewards, e.g., the MuJoCo-Maze (https://github.com/kngwyu/mujoco-maze) or Panda-Gym (https://github.com/qgallouedec/panda-gym)?
> >
> >
> > > title
> >
> > Besides, after reading other reviewers' comment, I also feel that the title could be more clear. For example, "Pink noise improves exploration in continuous control / deep reinforcement learning".

---

### Official Review · Reviewer_9xrD · 2022-10-23

**Confidence:** 4
**Correctness:** 3
**Technical Novelty And Significance:** 4
**Empirical Novelty And Significance:** 4
**Recommendation:** 8

**Clarity, Quality, Novelty And Reproducibility:**

### Clarity
The writing is overall clear and easy to follow, but there are several parts that can benefit from clearer writing or additional details:
- The introductory discussion on colored noise is unclear. For example, the discussion at the bottom of page 1 refers to the hyperparameter $\beta$ before Definition 1, which defines $\beta$ is provided. I recommend giving a concise, high-level explanation for colored noise upfront in the introduction, with an intuitive explanation for what $\beta$ controls.
- Overall, the introductory discussion on the connections between red noise and OU noise is quite confusing. At points, these two noise types seem to be treated synonymously, while at other points, these two noise types are explicitly framed as being similar, but different. To avoid this confusion, I recommend providing an intuitive definition of colored noise and their connection to OU noise in the introduction.
- Regarding the previous two points, which center on providing an upfront intuitive definition of colored noise and OU noise in the introduction, I suggest making Figure 4 the main Figure 1 for the paper, right on the first or second page. This figure does a great job of communicating an intuitive difference between the different choices of noise. The authors could also consider adding a fourth image for Brownian motion (red noise) for completeness.
- TD3/SAC/MPO are the main RL algorithms studied, but there is no detail about them provided anywhere in the paper. I suggest providing a high-level summary of these methods (highlighting how they are related) and pointing the reader to a more detailed synopsis of their implementation in the Appendix.
- The top-level column labels in Table 2 are confusing, and can be removed to benefit clarity.
- Figure 1 shows only 9 of 10 environments. Why not show all 10 environments for completeness?
- In Table 1.0, the authors could consider highlighting the "1.0 (Pink)" by changing the text color to pink for clarity.
- The authors do not define $\text{CN}_T(\beta)$ on Page 3. Though its definition can be guessed from context, its definition should be explicitly stated.
- The intuition for why temporally-correlated noise can improve exploration compared to white noise should be more clearly explained with the appropriate citation in the second paragraph (e.g. to Osband et al, 2016, which is cited elsewhere).
- The claim at the end of the second paragraph should include a citation to prior works that provide evidence for the claim: "Too much exploration is not beneficial for learning a good policy, as all algorithms require that the state-visitation distribution is approximately on-policy." This claim is not intuitively true, because the methods in question are all _off-policy_ methods.

### Quality
The presentation is polished with clearly labeled figures.

### Originality
This appears to be the first study of the impact of noise color in off-policy RL for continuous control.

### Reproducibility
The authors provide their implementation as a supplemental zip and clearly describe their experimental setup and hyperparameter choices. Thus the results in this study should be reproducible.

**Strength And Weaknesses:**

### Strengths
- The writing is overall quite clear. In particular, the exposition on colored noise was easy to digest and seemed quite comprehensive, especially combined with the discussion in the appendix.
- The experimental setup seems rigorous, covering a wide range of popular environments for continuous control, and making use of bootstrapped intervals and significance tests to assess the significance of their results.
- The results are strong, with clear, compelling evidence that pink noise should be used as the default noise choice for TD3/SAC/MPO as it consistency outperforms other choices, while being close in performance to the optimal noise choice when it is not the best choice.
- The toy environment experiments provide further compelling evidence that pink noise performs well across two isolated challenges that commonly occur in continuous control environments.

### Weaknesses
- The **major weakness** of this paper is the missing justification for the choice of $\sigma$ used in the baselines. It seems that $\sigma$ was set to a fixed value across all experiments, so it is possible that the advantages of pink noise disappear for other settings. To make the results fully convincing, it is necessary for the authors to show that the specific $\sigma$ used is the optimal $\sigma$ for the non-pink noise baselines, and show that replacing the noise type with pink noise for these values of $\sigma$ results in improved performance. If the authors can show this empirically or otherwise provide a sound argument for why the chosen $\sigma$ is the optimal choice for the baselines, I will raise my rating to an 8, as I believe the paper is otherwise very good.
- The authors assess only the mean returns of each method averaged over all time steps of training, across both SAC and MPO. This is certainly an unconventional metric. While it does track the area under the training curve, it is generally possible to have a large area under the curve with performance crashing later in training. Thus, these results should be supplemented with at least a measure of final performance or the training curves. The latter results are currently reported in the Appendix. The main results section can benefit from their inclusion.
- Results for the task-specific, best noise baseline (Section 4.2) is based on 10 seeds, while the pink noise baseline reports 20 seeds. Why is the oracle evaluation only based on the first 10 seeds? This seems like a potentially unfair comparison.
- Several section of the paper can benefit from refactoring of the content presented, additional citations, or generally clearer exposition (see the Clarity section of this review for more details)


**Summary Of The Paper:**

This paper studies the impact of the choice of colored noise in popular off-policy RL algorithms, TD3, SAC and MPO. They find that pink noise—an intermediary form of colored noise between totally, temporally uncorrelated white noise and temporally-correlated red noise (Brownian motion)—consistently results in the best performance, averaged over 10 different continuous control environments taken from several popular environment libraries. Pink noise tends either outperform other choices of noise, as well as a bandit baseline that adapts the key hyperparameter, $\beta$, controlling for the "color" of the noise, as well baselines that randomize or anneal the value of $\beta$ from red ($\beta=2$) to white noise ($\beta=0$).

**Summary Of The Review:**

Overall, this paper is very good. The authors provide a generally lucid account of the mechanics of colored noise and how it plays a role in modern, off-policy RL algorithms like TD3, SAC, and MPO. Their extensive experiments on standard continuous control benchmarks and toy environments provide compelling empirical evidence that pink noise is an ideal, default noise setting for these methods. This finding is significant, as it shows that the standard choices of white and OU noise are suboptimal. Using pink noise by default then provides a free improvement to this entire class of algorithms. This finding thus has significant impact on future work, which, by following the recommendation to use pink noise by default, should see generally improved performance on the problems to which these algorithms are applied. In turn, this improved performance may allow methods like TD3/SAC/MPO to be more viably applied to more challenging domains.

---

> ### Comment · Reviewer_9xrD · 2022-11-07
> **Updated review**
>
> After thinking more on this work, I found a hole in the empirical design of this work that makes me lower my initial rating of 8 to a 5. I updated the "Weaknesses" section of my original review with a description of this issue, which I also repeat below:
>
> > The major weakness of this paper is the missing justification for the choice of $\sigma$ used in the baselines. **It seems that $\sigma$ was set to a fixed value across all experiments, so it is possible that the advantages of pink noise disappear for other settings.** To make the results fully convincing, it is necessary for the authors to show that the specific $\sigma$ used is the optimal  for the non-pink noise baselines, and show that replacing the noise type with pink noise for these values of $\sigma$ results in improved performance. If the authors can show this empirically or otherwise provide a sound argument for why the chosen $\sigma$ is the optimal choice for the baselines, I will raise my rating to an 8, as I believe the paper is otherwise very good.

---

> > ### Author Response · Authors · 2022-11-08
> > **Clarification**
> >
> > Thank you very much for your highly detailed feedback. We will post a full response later, but want to clarify your latest comment about the noise scale $\sigma$ first.
> >
> > In the main text, we consider the algorithms MPO and SAC, both of which use a neural network to learn the action noise scale $\sigma(s)$ as a function of the state. In Equation (2) ($a_t = \mu(s_t) + \sigma(s) \odot \varepsilon_t$), we thus only change the random process from which $\varepsilon_t$ is drawn and keep the variance of $\varepsilon_t$ at 1 (as detailed in Appendix A, see e.g. Figure A.3). Of course it is possible to introduce an additional constant noise scale with which to multiply $\varepsilon_t$, but this should not be necessary, as the function $\sigma(s)$ is capable of learning such a constant factor itself.  Thus, we deliberately only change the temporal correlation of the action noise, which is one thing that the function $\sigma(s)$ cannot learn (as it is only a mapping from state to action noise scale), and keep the scale and shape (Gaussian) of the marginal distribution $p(\varepsilon_t)$ unchanged to the original SAC and MPO algorithms, where $\varepsilon_t \sim \mathcal N(0, 1)$.
> >
> > In Appendix B.1 we also report experiments on TD3, which does not learn the noise scale. Thus, for these experiments we do perform a grid search over $\sigma \in \{0.05, 0.1, 0.3, 0.5, 1\}$, and average the results across $\sigma$ (which shows a preference for pink noise).
> >
> > Do these clarifications resolve this issue, or did we misinterpret your comment?

---

> > > ### Comment · Reviewer_9xrD · 2022-11-08
> > > **Thank you for the detailed response**
> > >
> > > I thank the authors for the this detailed response, which addresses the concern I outlined above. I have thus reverted my score to 8.

---

> ### Author Response · Authors · 2022-11-17
> **Response to Reviewer 9xrD (1/2)**
>
> Thank you again for your detailed and helpful feedback. We are happy to read that you share our enthusiasm about our work.
>
> > The authors assess only the mean returns of each method averaged over all time steps of training, across both SAC and MPO. This is certainly an unconventional metric. While it does track the area under the training curve, it is generally possible to have a large area under the curve with performance crashing later in training. Thus, these results should be supplemented with at least a measure of final performance or the training curves. The latter results are currently reported in the Appendix. The main results section can benefit from their inclusion.
>
> We agree that learning curves and final policy performance are important metrics to make our results comparable to previous work. For this reason we have included learning curves and repeated the analysis of the average performance across tasks of Section 4.1 using the performance of the final policy instead of the average returns, all in Section B.2. Additionally, the average and final performance are compared on each individual environment in Section G.
> As these results agree with those presented in the main text, we find that including them in the appendix and referring to them from the main text is a good compromise to stay within the page limit.
>
> > Results for the task-specific, best noise baseline (Section 4.2) is based on 10 seeds, while the pink noise baseline reports 20 seeds. Why is the oracle evaluation only based on the first 10 seeds? This seems like a potentially unfair comparison.
>
> We ran experiments with each noise type using 20 random seeds. The results are used both to select the best-performing noise type on each environment, as well as to evaluate how well this noise type performs. As the best noise type selection is based on performance, it is necessary to evaluate the noise type separately from the runs where it was selected to avoid sampling bias (similarly to how a machine learning model is evaluated on a validation set separate from the training set). For this reason we select the oracle noise types on 10 seeds and evaluate on the 10 remaining seeds. We agree that using the first 10 seeds for selection is an arbitrary choice, so we have now repeated the analysis by selecting the noise type using the latter 10 seeds and evaluating on the first 10 seeds. This yields an almost identical outcome, and the combined results are now shown in an updated Figure 4.
>
> > The introductory discussion on colored noise is unclear. For example, the discussion at the bottom of page 1 refers to the hyperparameter $\beta$ before Definition 1, which defines $\beta$ is provided. I recommend giving a concise, high-level explanation for colored noise upfront in the introduction, with an intuitive explanation for what $\beta$ controls.
>
> We have expanded the explanation of $\beta$ in the introduction, and moved Figure 4 up from Section 6 to the first page, as you suggested. We hope this clarifies the discussion of temporal correlation and provides an intuitive explanation of colored noise.
>
> > Overall, the introductory discussion on the connections between red noise and OU noise is quite confusing. At points, these two noise types seem to be treated synonymously, while at other points, these two noise types are explicitly framed as being similar, but different. To avoid this confusion, I recommend providing an intuitive definition of colored noise and their connection to OU noise in the introduction.
>
> We agree that there was room for improvement. The main difference between OU noise and (stationary) red noise is that OU noise has a parameter ($\theta$) which controls the frequency cutoff in the PSD, where lower frequencies are treated like white noise ($\propto f^0$) and higher frequencies are treated like red noise ($\propto f^{-2}$). Red noise has no parameter and the complete PSD is $\propto f^{-2}$. In the main text we only consider the setting of $\theta = 0.15$, which is the most common setting if used as action noise in RL (e.g. it is the default in the StableBaselines3 and Tonic libraries we use, and is also recommended in the DDPG paper). This setting is in fact very similar to red noise (unless episode lengths become orders of magnitude longer), as can also be seen in the PSD plot Figure A.2.
> We have greatly extended our discussion of OU noise in Section A to highlight the differences between red noise and OU noise, but only consider $\theta = 0.15$ in the main text to aid clarity. Furthermore, we have added a sentence discussing $\theta$ to the main text to make the distinction clearer.

---

> > ### Author Response · Authors · 2022-11-17
> > **Response to Reviewer 9xrD (2/2)**
> >
> > > Regarding the previous two points, which center on providing an upfront intuitive definition of colored noise and OU noise in the introduction, I suggest making Figure 4 the main Figure 1 for the paper, right on the first or second page. This figure does a great job of communicating an intuitive difference between the different choices of noise. The authors could also consider adding a fourth image for Brownian motion (red noise) for completeness.
> >
> > Thank you for this excellent suggestion. We have moved this figure to page 1 and believe it will complement the intuitive description of the different noise types. We discusss red noise less deeply than OU noise because it is not a common choice that people use (unlike OU noise), and it does not outperform pink noise.
> >
> > > TD3/SAC/MPO are the main RL algorithms studied, but there is no detail about them provided anywhere in the paper. I suggest providing a high-level summary of these methods (highlighting how they are related) and pointing the reader to a more detailed synopsis of their implementation in the Appendix.
> >
> > We believe our method is largely independent of the undelying reinforcement learning algorithm for continuous control, and we have confirmed this empirically by demonstrating good performance on three different algorithms (MPO, SAC and TD3). Thus, the precise workings of these algorithms are not strictly relevant, so we have decided to reference the original papers, as well as the implementations we use, rather than reiterate these well-known methods.
> >
> > > The top-level column labels in Table 2 are confusing, and can be removed to benefit clarity.
> >
> > Thank you for the suggestion. We have removed the top-level column labels.
> >
> > > Figure 1 shows only 9 of 10 environments. Why not show all 10 environments for completeness?
> >
> > Two of the benchmark tasks we use for our experiments are part of the same "Cartpole" domain: the balance and swingup tasks. For clarity, we don't include the Cartpole screenshot twice in this figure.
> >
> > > In Table 1.0, the authors could consider highlighting the "1.0 (Pink)" by changing the text color to pink for clarity.
> >
> > Thank you for the suggestion. We have changed the text color to pink.
> >
> > > The authors do not define $\operatorname{CN}_T(\beta)$ on Page 3. Though its definition can be guessed from context, its definition should be explicitly stated.
> >
> > We have added a sentence to clarify the exact definition of $\operatorname{CN}_T(\beta)$ with a reference to the appropriate place in the appendix for more information.
> >
> > > The intuition for why temporally-correlated noise can improve exploration compared to white noise should be more clearly explained with the appropriate citation in the second paragraph (e.g. to Osband et al, 2016, which is cited elsewhere).
> >
> > We have added the suggested citation and think that the inclusion of the new "Figure 1" will also provide additional intuition about the role of temporal correlation.
> >
> > > The claim at the end of the second paragraph should include a citation to prior works that provide evidence for the claim: "Too much exploration is not beneficial for learning a good policy, as all algorithms require that the state-visitation distribution is approximately on-policy." This claim is not intuitively true, because the methods in question are all off-policy methods.
> >
> > The reason why the state-visitation distribution of off-policy rollouts must cover the on-policy states is due to the fundamental nature of statistical learning: the sampling distributions (of states) must be the same during "training" (exploratory, off-policy rollouts) and "testing" (on-policy evaluation rollouts). We have changed the wording in this sentence to clarify this point.

---

> > > ### Comment · Reviewer_9xrD · 2022-11-19
> > > **Response to authors**
> > >
> > > Thanks to the authors for the detailed comments, and I am glad to see their incorporating some of my suggestions as improvements to the paper.
> > >
> > > Given that my questions have been addressed and that I find this paper overall well-written and its findings useful to future work, I will keep my current rating for this paper.

---

### Official Review · Reviewer_D8U1 · 2022-10-25

**Confidence:** 4
**Correctness:** 4
**Technical Novelty And Significance:** 3
**Empirical Novelty And Significance:** 3
**Recommendation:** 8

**Clarity, Quality, Novelty And Reproducibility:**

**Clarity**

Exceptionally good clarity. My only comment is that it may be good to say that $f$ is the frequency in Definition 1.

**Quality**

The quality of the experimental analysis was very high.

**Novelty**

The novelty is incremental.

**Reproducibility**

I believe the work is reproducible as sufficient details were provided. They also said that they will release the code.

**Strength And Weaknesses:**

**Strengths:**
- The experimental work was solid and thorough doing a proper statistical analysis of the results. I believe the results are convincing.
- Noise is added into most RL algorithms, so this work is applicable to a wide range of researches.
- The paper was well written and included interesting explanations.

**Weaknesses:**
- If I have to name a weakness, I would say that ultimately the type of noise is an unalluring component of the RL system, and the specific type does not seem to make or break the performance of the algorithm (e.g., in Figure B.4 the other noise types also perform mostly fine). Even though pink noise statistically performs better, the improvement is not a conceptual advance in terms of performance.
- I didn’t like the title and I would recommend changing it (this doesn't affect the review).

**Summary Of The Paper:**

The work presents a thorough investigation of the type of used action noise in reinforcement learning.

The typically used noise types are either Gaussian noise or Ornstein-Uhlenbeck noise (OU), which is temporally correlated.
These two noise types belong to a family of colored noise, where Gaussian noise is white noise and OU noise is red noise.
The family of noise has a frequency spectrum proportional to $f^{-\beta}$, where $\beta=0$ corresponds to white noise (a flat frequency spectrum) and $\beta = 2$ corresponds to OU noise (the frequency power spectrum decays). In the work they consider other noise types with $0 < \beta < 2$. In particular, they determined that $\beta=1$ (pink noise) performs well as a default setting.
The noise can be generated using an FFT based algorithm prior to starting the episode, stored into a vector and then selected from the vector (this is a sensible procedure and is often also used for other noise types).

They performed experiments across 10 different tasks taken from DMControl Suite, OpenAI Gym and Adroit hand Suite. They tested different noise types for the SAC, MPO and TD3 algorithms. Experiments were run with 20 seeds.
To aggregate the performance across tasks, they normalized the performance in each task to have 0 mean and variance 1.
Pink noise had the best average performance across all tasks. Moreover, pink noise achieved the highest score in 3/10 tasks and the difference to the best was statistically insignificant in 5/10 of the remaining tasks. Meaning that pink noise was comparable to the best in 8/10 tasks. The difference in performance for pink noise in the remaining two tasks was also low, and the work included much discussion and further analysis of these results.

In addition the work tested scheduling the noise color schedule, as well as selecting the noise type using a bandit algorithm. Neither of these was significantly better than simply using a fixed $\beta=1$ pink noise.

Section 6 included an intuitive explanation of why pink noise might be better than other noise types, e.g., Figure 4 showed that in a random walk, Gaussian noise does not disperse much from the center, OU noise quickly leaves the center and gets stuck in the edges of the environment, while pink disperses from the center while traveling through a greater range of the intermediate distances.

The analysis indicated that pink noise is a better default setting for the noise compared to the previous Gaussian and OU noises.

**Summary Of The Review:**

This is a well written paper on a topic that is relevant to many RL researches. The experimental work was thorough, and the discussion was interesting. Pink noise may become a default noise setting for future RL algorithms. I think this paper is a clear accept.

**Update**
I thank the authors for their response. I believe my assessment remains adequate, and I am keeping the score.

---

> ### Author Response · Authors · 2022-11-17
> **Response to Reviewer D8U1**
>
> Thank you very much for your positive review!
>
> > If I have to name a weakness, I would say that ultimately the type of noise is an unalluring component of the RL system, and the specific type does not seem to make or break the performance of the algorithm (e.g., in Figure B.4 the other noise types also perform mostly fine). Even though pink noise statistically performs better, the improvement is not a conceptual advance in terms of performance.
>
> In many cases this is true, but it seems to us that the noise type can indeed make a drastic difference on certain environments. For example, the MountainCar and Hopper environments are not solved at all by MPO with the default white noise.
>
> > I didn’t like the title and I would recommend changing it (this doesn't affect the review).
>
> Our experiments, especially the "oracle" comparison in Section 4.2 indicate that using pink noise performs as well as a large hyperparameter search for the noise type. In this sense, it is "all you need". Alternative titles appeared less descriptive to us, but we are open to suggestions.
>
> > it may be good to say that $f$ is the frequency in Definition 1
>
> Thank you for spotting this, we have included it in the definition now.

---

### Author Response · Authors · 2022-11-18
**General Response**

We want to thank all three reviewers for their time in reading our paper, as well as for the positive response and valuable comments. Based on this feedback, we have been able to implement several changes, resulting, in our opinion, in a greatly improved paper.

Changes:
- We ran additional experiments with Ornstein-Uhlenbeck noise of corrected noise scale $\sigma$, such that the marginal distributions $p(\varepsilon_t)$ approach $\mathcal N(0, 1)$, to make the comparison with colored noise and white noise more fair.
    - The results of these experiments are very close to the previously reported experiments of noise scale $\sigma = 1$.
    - The derivation of the noise scale correction is included in Section A.1
    - The previous OU experiments with $\sigma = 1$ are included in Section B, with the new experiments taking the place of the previous ones everywhere in the main text.
    - These new experiments affect the performance normalization constants and the oracle/anti-oracle, meaning that many numbers in the text and tables change slightly. However, all conclusions that are drawn from these numbers remain valid and unaffected.
- We have included an extensive discussion of the hyperparameters $\theta$ and $\Delta t$ of Ornstein-Uhlenbeck noise and how these are related to the $\beta$ parameter of colored noise (Section A.2).
    - We have analyzed the behavior of OU noise of different parameterizations on the bounded integrator and harmonic oscillator environments of Section 6.
    - The results show that pink noise surpasses all parameterizations of OU noise in terms of generality (worst-case performance), thereby strengthening our points from Section 6 and pink noise's claim to being a good default action noise.
- Throughout the text we have included minor clarifications and corrections as suggested by the reviewers
    - We have corrected Figure A.3

All changes to the text are highlighted in blue in the paper revision.

---

### Decision · Program_Chairs · 2023-01-20

**Decision:**

Accept: notable-top-25%

**Justification For Why Not Higher Score:**

See weaknesses section. Interpolating between known, heuristic, exploration methods is useful. It can be enlightening. But there is a pretty high bar for oral presentations and reaching it likely requires more substantial new thinking.

**Justification For Why Not Lower Score:**

The paper is very well executed and offers some practical guidance. The reviewers are unanimous in feeling the paper deserves clear acceptance.

**Metareview: Summary, Strengths And Weaknesses:**

This paper studies RL in control problems with continuous action spaces. A common default is to generate exploration by perturbing selected actions by random Gaussian noise that is chosen independently across time (White noise). This kind of noise fails to generate 'deep exploration' that is required in many problems. Lillicrap et. al [2016] inject action noise generated according to an Ornstein-Uhlenbeck process (red noise), which can force the agent to persistently perturb actions in a particular direction. The current paper studies a parameterized class of noise distributions that essentially interpolates between these cases. A particular choice of 'pink noise' is found empirically to offer better default across ten environments than either extreme.

*Strengths*: While the use of temporally correlated noise is not new, the authors have udnerstood the importance of this design choice, evaluated it clearly, and offered intuition on when it is most important. It terms of new methodology, pink noise does, indeed, appear to be a nice default. Pragmatically speaking, when don't understand who to reach a scientific ideal (i.e. systematic exploration), it is helpful to have a few tuning parameters. The beta parameter that interpolates smoothly between white noise (beta=0), pink noise (beta =1) and colored noise (beta=2), could really help a practitioner.

All reviewers felt the paper was clear accept.

*Weaknesses:*  Adding temporally correlated noise is not, in general, sufficient to generate efficient exploration and overcome the deficiencies of white noise. It would be helpful if the paper offered some discussion of when temporally correlated noise might be enough, rather than simply tuning the degree of correlation. The paper focuses on an extremely narrow range of exploration methods.

**Note From Pc:**

if the above contains the word "oral" or "spotlight" please see: "oral" presentation means -> notable-top-5% and "spotlight" means -> notable-top-25%. As stated in our emails, we are disassociating presentation type from AC recommendations